# The Bats of Greece: An Updated Review of Their Distribution, Ecology and Conservation

**DOI:** 10.3390/ani13152529

**Published:** 2023-08-05

**Authors:** Panagiotis Georgiakakis, Artemis Kafkaletou Diez, Ioanna Salvarina, Petr Benda, Geoffrey Billington, Christian Dietz, Jacquie Billington, David Cove, Stephen Davison, Martyn Cooke, Eleni Papadatou

**Affiliations:** 1Natural History Museum of Crete, University of Crete, Knossos Ave., 71409 Irakleio, Greece; 2Independent Researcher, Andrianaki N. 11, 73133 Chania, Greece; 3Independent Researcher, Them. Sofouli 4, 54646 Thessaloniki, Greece; 4National Museum, Václavské Nám. 68, 115 79 Praha, Czech Republic; 5Faculty of Science, Charles University, Viničná 7, 128 00 Praha, Czech Republic; 6Greena Ecological Consultancy, Stonehaven BA11 5HH, UK; 7Biologische Gutachten Dietz, Balinger Strasse 15, 72401 Haigerloch, Germany; 8Independent Researcher, 26 Parkfield Crescent, Ruislip HA4 0RE, UK; 9Independent Researcher, Newport NP10 9JJ, UK; 10Independent Researcher, Charlwood RH6 0DR, UK; 11Independent Researcher, Huddersfield HD7 6BW, UK

**Keywords:** Chiroptera, biodiversity, roosting ecology, foraging, Balkans, Mediterranean

## Abstract

**Simple Summary:**

We review and summarise all published and unpublished information currently available on the distribution and ecology of the bats of Greece and provide updated distribution maps of all bat species in the country. Additionally, we provide information on the bat species ecology, in terms of roost and foraging/commuting habitat use, altitudinal distribution, winter activity and landscape characteristics around major roosts. Finally, we highlight the current research and conservation needs for the bats of Greece.

**Abstract:**

Bats of Greece have been studied since the second half of the 19th century. Their distribution and ecology, however, remain poorly understood. Conservation efforts for the protection of the roosting and foraging habitats of their populations in Greece are limited. To date, 35 bat species have been recorded from Greece. Four species (*Eptesicus anatolicus*, *Plecotus auritus*, *Myotis brandtii* and *Rousettus aegyptiacus*) have a limited distribution in the country and the presence of one species (*Myotis mystacinus*) requires verification. The present study summarises all existing knowledge and adds several hundred new records on the distribution of bats of Greece. Additionally, it provides a summary of new insights on various aspects of their roosting ecology, foraging habitat use, altitudinal distribution, winter activity and landscape characteristics around major roosts. Finally, it discusses the current research and conservation needs of Greek bats.

## 1. Introduction

### 1.1. The Fauna of Greece

Greece has a rich and diverse fauna, with more than 30,000 species described from several phyla [1,2]. The Greek fauna includes typical southeastern Mediterranean species, but also species of Asiatic and/or African origin, with different distribution patterns across the country. Paleoclimatic phenomena (e.g., ice ages and glaciations of the Pleistocene, Messinian salinity crisis of the Late Miocene), the petrological composition and geological history of the Aegean area [3] and the resulting current variability in landscape relief, vegetation composition and climatic patterns are the main drivers of such biological diversity across the country [4]. The prevalence of carbonic rocks with many mountains consisting of karstified limestone has provided roosts to many bat species, while the relict forests of the mountainous areas and the numerous gorges and wetlands of variable sizes constitute excellent feeding grounds for most bats [3].

### 1.2. Bat Research in Greece

Bats were first recorded in Greece in the second half of the 19th century by Lindermayer [5], who reported 10 bat species from the island of Euboea. By the mid-1960s, 25 bat species had been recorded by non-native researchers in various parts of the country; their findings are summarised in Kanelli and Hadzisarantou [6] and Ondrias [7]. Iliopoulou-Georgudaki was the first Greek researcher who systematically studied bats in the country, focusing on their taxonomy and systematics [8]. Her resulting data are summarised in Hanák et al. [9]. Frequent research trips by Professor O. von Helversen (University of Erlangen, Germany) and his students over the course of ca. 50 years (1958–2007) contributed significantly to the increase in the knowledge on the distribution, systematics and ecology of bats in Greece partly summarised by [9,10,11,12]. The vast majority of these data remained unpublished; the main authors of this paper, however, were granted access to a version of his database following his passing and a large number of records have been included in the present work. A significant amount of data on the bats of Greece were also collected by Czech researchers (Charles University, Prague) during mammal surveys carried out in the 1980s and 1990s [9]. By the end of the 20th century, the number of known bat species in the country had been raised to 32 (all records reviewed and summarised in Hanák et al. [9]).

From the 2000s, young native scientists were involved in surveys and research on the distribution, ecology and conservation of bats of Greece in the framework of their postgraduate or doctoral dissertations or other research activities. The composition and ecology of bats in the prefecture of Evros, northeastern Greece and Crete were extensively studied by Papadatou [13,14,15,16,17] and Georgiakakis [18,19,20,21,22,23], respectively.

The bats of the National Park of Prespa in northwest Greece and adjacent areas in Albania and North Macedonia, were the subject of detailed bat surveys between 2004 and 2011, initiated and assisted by the Groupe Mammalogique Breton, and involving local researchers and surveyors from all three countries [24]. These surveys resulted in a transboundary action plan for the conservation of bats in the area, supported by the Society for the Protection of Prespa in Greece [24], and other publications [25,26,27]. Further information on bats of the National Park and its surroundings was later collected by Uhrin et al. [28].

More information on the distribution and ecology of bats was also collected by Greek researchers within the framework of various projects in the Administrative Regions of West Macedonia, Central Macedonia and of East Macedonia and Thrace, e.g., [29,30,31,32,33,34,35,36,37]; Thessaly [38], Central Greece, e.g., [39], West Greece and Peloponnese, e.g., [40,41,42,43,44]; Epirus, e.g., [45,46]; and islands in the Ionian, e.g., [47,48] and in the Aegean Sea, e.g., [49,50]. Non-native researchers further contributed to the collection of a significant amount of data from various parts of Greece [51,52,53,54,55,56,57].

From 2011, bat surveys have been organised on a number of Greek islands during regular summer and/or autumn expeditions by Greena Ecological Consultancy (based in the U.K.) in collaboration with the Natural History Museum of Crete and Greek researchers and volunteers. The data collected are presented here for the first time.

Μany field records of bats were collected by the main authors of this paper in the framework of monitoring and conservation status evaluation programmes, and conservation and management projects, within the framework of the Habitats Directive 92/43/EE [58,59,60,61,62,63], as well as impact assessment projects for wind farms and other constructions [64,65,66]. The vast majority of these records have only been published in respective reports and a small proportion were presented at conferences, e.g., [40,67,68]. More recently, records on cave bats were collected within the framework of the project LIFE GRECABAT-LIFE17 NAT/GR/000522 [69]. Finally, information has been collected through observations inside roosts by cavers or echolocation call recordings of bats in flight by volunteers.

In this context, the present review aims to summarise all published and unpublished information currently available on the distribution and ecology of the bats of Greece and to provide: (a) updated distribution maps of all bat species in the country, including the mainland and islands for which bat presence is known, (b) information on their ecology, in terms of roost and foraging/commuting habitat use, altitudinal distribution, winter activity and landscape characteristics around major roosts, and (c) highlights on their current research and conservation needs.

## 2. Materials and Methods

All available, both published and unpublished, data on particular bat species occurrence, colony size, etc., in Greece are stored in the National Database of the Chiroptera of Greece (NDCG), hosted by the Natural History Museum of Crete, University of Crete, Irakleio. A simlyfied version of the database is provided in the Appendix A, where site names and geographic information is given and main roosts and original data are noted. The database, featuring information from more than 1000 roosts and 2000 foraging/commuting sites was used to create distribution maps for all bat species and to summarise information on their habitat use and ecology in the country. Data were collected using various methods, including echolocation call recordings of bats in flight, observations inside roosts, captures at/near roost entrances and at foraging/commuting grounds, as well as radiotelemetry.

Many records from echolocation call recordings were regarded as tentatively identified and were not used, specifically of species whose echolocation call values overlap, unless their presence was verified through other methods (e.g., confirmed through individuals identified in hand and species specific social calls present in sound recordings); these include species in the genera *Nyctalus*, *Pipistrellus*, *Myotis*, *Plecotus* and *Miniopterus* [15,70,71]. Where such a record potentially affects species distribution, it is discussed in the respective species account.

Distribution maps were created, and roosting and foraging habitat use was assessed using ArcGIS Pro (3.1). Habitat use outside roosts (foraging and commuting habitats combined) was assessed for species for which data were deemed adequate, using the Join tool to assign each record point to a certain land cover class (Corine Land Cover 2018). Similarly, to quantify the landscape composition around known major roosts, i.e., roosts that hosted more than 100 individuals on average, we created a 5 km buffer zone around each roost (ca. the average distance to hunting grounds for cave species) and estimated the average proportion of each land cover class (Corine Land Cover 2018) for the sum of selected roosts for each species.

Distribution maps and information presented per species may be biased for several reasons. First, uneven sampling effort; for example, several hundred of small islands and islets have been inadequately sampled or not sampled at all for logistic reasons. Only 20 islands were thoroughly investigated and from 27 islands only limited presence data are available (Figure 1). Limited or no sampling has taken place in many areas of the mainland, either because they are highly inaccessible (for example, rocky cliffs on high mountains and steep shores) or severely degraded (e.g., intensively cultivated plains). Second, sampling methods and the ease in species detection and identification; for example, species roosting in accessible buildings and underground sites are easier to locate and identify than those roosting in trees/rocks in extensive woodland on mountainous areas. Similarly, the distribution and abundance of species that have louder and more easily identified echolocation and social calls may be over-represented in sample recordings compared to species with quieter calls that, moreover, overlap with those of other species. Third, the lack of long-term systematic studies and monitoring. With a few exceptions, for the majority of species in most of the country, information has been collected opportunistically and many areas have not been surveyed for many years. This means that the current distribution range of some species and the size of several colonies may have changed.

For the altitudinal distribution of bat species, we estimated the minimum and maximum elevation (in metres above sea level—m a.s.l.) and the respective mean and median values from all records of presence. We also estimated the minimum and maximum elevation of all known roosts.

For species names, we followed the taxonomy proposed by Hackländer and Zachos [72].

## 3. Results—Species Accounts

### 3.1. Rhinolophus ferrumequinum (Schreber, 1774)

The greater horseshoe bat *Rhinolophus ferrumequinum* (Schreber, 1774) was first recorded in Greece in 1855 by Lindermayer [5]. It is one of the most common bat species in Greece (Figure 2, Table 1), since it has been found in several hundred locations in mainland Greece and on 22 islands of the Aegean and the Ionian Seas, the smallest being Kastellorizo (ca. 12 km^2^) in the southeast of the Aegean Sea.

Τhe assignment of the Cretan population to a separate subspecies (*Rhinolophus ferrumequinum creticum*) suggested by Iliopoulou-Georgudaki and Ondrias [73] was not supported by the results of subsequent analyses of biometric traits, echolocation call recordings and genetic indices [18] and references therein.

Single individuals or colonies of variable sizes were found in 337 roosting sites, both underground and overground, such as caves (including sea caves), disused mines, concrete tunnels and buildings. The 28 largest colonies consisted of between 100 and 525 individuals; most of these were nursery colonies found in caves. Nursery colonies of equal size, were also found in buildings such as old stone chapels and abandoned houses. Some underground roosts (caves, mines, concrete tunnels) were occupied even in winter by variable numbers of individuals (up to a few hundred). In winter, *R. ferrumequinum* has been observed to emerge from its roosts, on warm and calm nights with little or no wind and rain. Outside roοsts, the species was usually found in mosaics of agricultural areas, woodland/forests and shrublands.

The earliest record of births comes from Crete and was in late May; births may start later in northern Greece, since timing of births is affected by ambient temperatures and, thus, latitude [71]. The species has been encountered from the sea level (0 m a.s.l.) up to 1870 m a.s.l. The known roost elevation ranges from sea caves to 1500 m a.s.l. The main roosts of the species are surrounded (on a 5 km radius) mainly by sclerophyllous vegetation, natural grassland, olive groves, water bodies and non-irrigated arable land.

### 3.2. Rhinolophus hipposideros (André, 1797)

The lesser horseshoe bat *Rhinolophus hipposideros* (André, 1797) was first recorded in Greece in 1855 by Lindermayer [5]. It is one of the most common bat species in Greece (Figure 3), since it was found in several hundred locations in mainland Greece and on 19 islands of the Aegean and the Ionian Seas, the smallest being Ithaki (ca. 96 km^2^) in the Ionian Sea. Single individuals or colonies of variable sizes were found in 320 roosting sites, both underground and overground, such as caves (including sea caves), disused mines and tunnels, buildings and, on a few occasions, also bridges and trees (von Helversen, unpublished data). The species may roost in underground sites all year round, but all large nursery colonies were in buildings. The 73 largest colonies consisted of between 10 and 200 individuals. On warm and calm winter nights, the species may emerge to forage (own observations). Outside roοsts the species was usually encountered in mosaics of semi-natural areas, broadleaf woodland/forests, tree cultivations, agricultural areas, pastures, meadows and water surfaces.

The earliest record of births comes from Crete and was in late May, probably starting later in the north of the country. The species has been encountered from the sea level up to 1700 m a.s.l. (A. Alberdi, pers. comm.). The known roost elevation ranges from sea caves to 1510 m a.s.l. The main roosts of the species are surrounded mainly by broad-leaved forest, transitional woodland shrub, mixed forest and water bodies.

### 3.3. Rhinolophus euryale Blasius, 1853

The Mediterranean horseshoe bat *Rhinolophus euryale* Blasius, 1853 was first recorded in Greece in 1926 by Bolkay [74]. It is widespread in mainland Greece and has also been found on eight islands of the Aegean and the Ionian Sea, the smallest being Lefkada (ca. 303 km^2^) in the Ionian Sea (Figure 4), but it is absent from Crete. Single individuals or colonies of variable sizes were found roosting in 77 underground sites, such as caves (including sea caves), disused mines and concrete tunnels. The 10 largest colonies (including nurseries) consisted of 100 to 500 individuals. Some underground roosts are occupied also in winter, by fluctuating numbers of individuals (up to few hundred). On warm and calm winter nights, the species has been observed emerging from its roosts. Outside roοsts, the species was usually found in woodland/forests, semi-natural areas and tree cultivations.

The earliest record of lactating females comes from Lesvos and it was in the first week of June. The species has been encountered from the sea level up to 1388 m a.s.l. (the same is valid for roosts). The main roosts of the species are surrounded mainly by mixed forest, sclerophyllous vegetation, natural grassland and sea.

### 3.4. Rhinolophus mehelyi Matschie, 1901

Mehelyi’s horseshoe bat *Rhinolophus mehelyi* Matschie, 1901 was first recorded in Greece in 1955 by Strinati [75]. It has a patchy distribution in Greece and it is relatively rare (Figure 5) as in Albania [57], North Macedonia [76], but it is more common in Bulgaria [77] and Turkey [78]. The species has only been found in 22 locations in mainland Greece from the regions of East Macedonia and Thrace to Central Greece, West Greece and the Peloponnese. *R. mehelyi* was also found in three localities on the island of Lesvos (in the north of the Aegean Sea) very close to the Anatolian coast; a cave (in early June 2017 and early May 2015, small numbers of individuals including pregnant and lactating females), an ancient mine in mid-September 2000 (one male) [9] and a wetland in early September 2002 (one individual-to date the only record outside roosts; Alivizatos and Dimaki, pers. comm.). The small number of observations on the island may indicate seasonal migration from the Turkish population but only systematic research could confirm this hypothesis. Single individuals or colonies have been found in 19 underground sites in total in the country, including caves and disused mines and tunnels. The six largest colonies (including nursery roosts) consisted of 20 to 500 individuals. Only a handful of winter colonies are known, all in caves. On warm and calm winter nights bats may emerge to forage.

*Rhinolophus mehelyi* has been encountered from the sea level, up to 890 m a.s.l. (the same is valid for roosts). The main roosts of the species are surrounded mainly by non-irrigated arable land, complex cultivation patterns, sea and permanently irrigated land.

### 3.5. Rhinolophus blasii Peters, 1866

Blasius’ horseshoe bat *Rhinolophus blasii* Peters, 1866 was first recorded in Greece in 1855 by Lindermayer [5]. It is relatively common in Greece, since it has been found in many locations in mainland Greece and on 24 islands of the Aegean and the Ionian Seas, the smallest being Kastellorizo (ca. 12 km^2^) in the southeast of the Aegean Sea (Figure 6). Single individuals or colonies of variable sizes were found roosting in 120 underground sites, mostly caves, but also disused mines and concrete tunnels. The 16 largest colonies consist of between 100 up to a few hundred individuals. The largest nursery colonies were found in caves. Some roosts are occupied even in winter, by fluctuating numbers of individuals (up to a few tens). On warm and calm winter nights, the species has been observed to emerge from its roosts. Outside roosts, the species was usually found in woodland/forests, semi-natural areas, tree cultivations and natural areas with low vegetation.

*Rhinolophus blasii* has been encountered from the sea level up to 1500 m a.s.l. The known roost elevation ranges from sea level to 1100 m a.s.l. The main roosts of the species are surrounded mainly by sclerophyllous vegetation, natural grassland, non-irrigated arable land, complex cultivation patterns and land principally occupied by agriculture, with significant areas of natural vegetation and broad-leaved forest.

### 3.6. Myotis blythii (Tomes, 1857)

The lesser mouse-eared bat *Myotis blythii* (Tomes, 1857) (=*oxygnathus* (Monticelli, 1885)) was first recorded in Greece in 1881 by Winge [79]. It is a common and abundant bat species in Greece, since it has been found in tens of roosts, foraging and commuting sites in mainland Greece and on 17 islands of the Aegean and Ionian Sea (Figure 7). The smallest island where the species was found is Ithaki (ca. 96 km^2^) in the Ionian Sea. In most of mainland Greece and on some islands, the species occurs in sympatry with the morphologically similar *Myotis myotis*. Identification of the two species is based on upper tooth row length and other morphological characters [71]. Some individuals captured in the region of East Macedonia and Thrace, Central Macedonia and on North Aegean islands could not be assigned to either species [80], perhaps indicating hybridization, as it has been documented elsewhere in eastern Europe [81].

It was proposed that the Cretan population belongs to the subspecies *M. blythii omari* Thomas, 1905 [82,83,84,85], but subsequent studies showed that it was dimensionally intermediate between the two continental forms *M. b. oxygnathus* and *M. b. omari.* Τhe masticatory apparatus is significantly smaller, indicating a taxonomic transition or the influence of the island environment and diet specialisation [18,21]. More recently, it was proposed that *M. b. oxygnathus* is a separate valid species in Europe (as *M. oxygnathus*) [12,86,87], but this view was not widely accepted [18,57,88].

Approximately 130 roosts of solitary individuals or colonies of variable sizes were known in Greece. The 24 largest colonies consisted of between 100 and ca. 2400 individuals and most of them were nursery colonies. All nursery and winter roosts were found in caves (including sea caves), and disused mines and tunnels, in contrast to the species northern distribution where nursery roosts are in buildings [71]. Bridges and buildings were rarely used, by small groups or solitary males. Singly roosting individuals were also found in rock crevices in rock overhangs.

Outside roosts, *M. blythii* was mostly recorded in semi-natural areas, woodland edge/forests and shrublands. The earliest record of births comes from Crete and was in the last week of May, but lactating females were encountered in late July in other parts of the country. The species was recorded from the sea level (0 m a.s.l.) up to 2085 m a.s.l. (a single individual torpid in a vertical shaft in western Crete). Current data suggest that, in winter, the species is not usually found in low elevation areas. Four hibernacula were found above 1250 m a.s.l. in Crete and one on the island of Kefalonia at 1388 m a.s.l., hosting up to 100 individuals each. The main roosts of the species are surrounded mainly by sclerophyllous vegetation, sea and ocean, olive groves, land principally occupied by agriculture, with significant areas of natural vegetation and complex cultivation patterns.

### 3.7. Myotis myotis (Borkhausen, 1797)

The greater mouse-eared bat *Myotis myotis* (Borkhausen, 1797) was first recorded in Greece in 1839 by Keyserling and Blasius [89]. It appears to be less common than its sibling species, *Myotis blythii* and it is absent from the islands of the south Aegean Sea, including Crete, where the latter species is common. *M. myotis* has been found in several tens of localities in mainland Greece and only on six islands of the Aegean and the Ionian Seas, the smallest being Lefkada (ca. 303 km^2^) in the Ionian Sea (Figure 8). The species often forms mixed colonies with *M. blythii*. Single individuals or colonies of variable sizes were found in 55 roosting sites, including caves, disused mines and railway tunnels, bridges and buildings, with seven of them hosting the largest colonies of the species in Greece, between 100 and 1500 individuals, all in underground sites. These numbers, however, are rough estimates and should be treated with caution, because the species often roosts with the morphologically similar *M. blythii.* Unlike northern and central Europe, in Greece *M. myotis* forms its nursery colonies in underground sites, whereas single individuals (most often males, usually in summer) roost in buildings and crevices in bridges or rock overhangs. Νo winter roosts are currently known. Outside roosts, the species was found mostly in semi-natural areas, grasslands and woodland/forests.

*Myotis myotis* has been encountered from 5 m a.s.l., up to 1950 m a.s.l. (A. Alberdi, pers. comm.). Known roost elevation ranges from 5 m to 1550 m m.a.s.l. The main roosts of the species are surrounded mainly by sclerophyllous vegetation, permanently irrigated land, complex cultivation patterns, non-irrigated arable land and transitional woodland-shrub.

### 3.8. Myotis alcathoe von Helversen and Heller, 2001

Alcathoe’s whiskered bat *Myotis alcathoe* von Helversen and Heller, 2001 appears to have a limited and scattered distribution in Greece (Figure 9). It is also rare in Albania [57], North Macedonia [90], Bulgaria [77] and Turkey [91]. *M. alcathoe* has been found in 12 localities in mainland Greece and none of the islands. The species was first described in 2001 by von Helversen et al. [92] cf. [93], from Greece; a single colony was found in a plane tree (*Platanus orientalis*), at ca. 700 m a.s.l. [92], consisting of three adult females and two juveniles. Outside roosts, it was trapped above streams with broad-leaved trees, mostly in semi-natural areas and broadleaf forests. The species has been encountered from the lowlands (285 m a.s.l.) up to 930 m a.s.l. 

### 3.9. Myotis davidii (Peters, 1869)

David’s myotis *Myotis davidii* (Peters, 1869) appears to be common in Greece. Ιt has been found in many locations in mainland Greece and on 10 islands of the Aegean and the Ionian Seas, the smallest being Ikaria (ca. 255 km^2^) in the East Aegean Sea (Figure 10). Until recently, the name *Myotis aurascens* (Kuzâkin, 1935) was used instead, but it has now been replaced by the widely accepted *M. davidii* [94]. Most older records of *Myotis mystacinus* probably belong to *M. davidi* [9,18,93,95,96]. Eight roosting sites have been located, in bridges and buildings, but individuals have also been trapped at the entrance of several caves on islands (mostly in Crete), on their way in, from April to October (except June). The only confirmed maternity colony was found in Central Macedonia where at least five pregnant and lactating females were found in the crevices of a bridge (mid-June 2015). Outside roosts, *M. davidii* was found mostly in semi-natural areas, broadleaf woodland/forests and open areas with low vegetation. The species has been encountered from the sea level, up to 1950 m a.s.l. (A. Alberdi, pers. comm.). The known roost elevation ranges from 5 m to 1500 m a.s.l. 

### 3.10. Myotis brandtii (Eversmann, 1845)

Brandt’s bat *Myotis brandtii* (Eversmann, 1845) is likely the rarest bat species in Greece (Figure 5) and it is also rare in Albania [57], Bulgaria [77] and Turkey [91]; it has not yet been documented in North Macedonia. *M. brandtii* was first recorded in Greece in 2001 by von Helversen [12]. To date, it has only been found in four locations in the mountains of Northern Pindos and the Prespa Lakes in northwest Greece ([12,24,97]; own unpublished data). These findings suggest that the species distribution is likely confined to the north of the country, which is likely the southern limit of its European distribution. The capture of a juvenile bat in flight in late June suggests that the species breeds in the country. Brandt’s bat has been encountered in medium to high elevation areas; from 1276 m up to 1700 m a.s.l. The species was captured at creeks within forested areas, showing a preference for mountainous broadleaf forests.

### 3.11. Myotis capaccinii (Bonaparte, 1837)

The long-fingered bat *Myotis capaccinii* (Bonaparte, 1837) was first recorded in Greece in 1955 by Lindberg [98] and Aellen [99]. It has been found in several locations on mainland Greece and on nine islands of the Aegean and the Ionian Seas, the smallest being Thassos (ca. 380 km^2^) in the North Aegean Sea (Figure 11). Colonies of variable sizes or single individuals have been found in 54 roosts; the vast majority of these were underground sites such as caves, and disused mines and tunnels. Most hibernating and nursery colonies were found in caves and disused mines, with the exception of a winter colony of 20 individuals observed in an ancient castle in Central Macedonia, among the stones of the arc-vault in December 2013. Bats were observed emerging to forage and exhibiting swarming behaviour as late as in December in the Peloponnese [41]. In Thrace, northeastern Greece, a systematic study showed that the species used a network of different roosts (caves, disused mines) within and between seasons; bats also moved between Turkey, Greece and Bulgaria across seasons [16]. This is in conjunction with the results of studies in France, where the species was also shown to use a network of roosting sites within and between seasons [100]. It is highly likely that the same behaviour is exhibited by this species in other areas of Greece and by other cave-dwelling species. The 12 largest known colonies consisted of between 100 to over 600 individuals, usually mixed with other cave-dwelling bats, such as *Miniopterus schreibersii*, *Myotis blythii* and *Rhinolophus* species.

Outside roosts, *M. capaccinii* was encountered mostly in cultivated and semi-natural areas, woodland/forests and wetlands, but generally over water surfaces, such as ponds, lakes and rivers, and even the sea ([13] and unpublished data from various parts of Greece). The earliest record of volant young comes from Crete and it was in the second week of June, whereas in Thrace, northeast Greece, they appeared in early July [14]. The species has been encountered from the sea level up to 1120 m a.s.l. including both roosts and foraging/commuting sites. The main roosts of the species are surrounded mainly by mixed forest, broad-leaved forest, natural grassland, non-irrigated arable land and complex cultivation patterns.

### 3.12. Myotis daubentonii (Kuhl, 1817)

Daubenton’s bat *Myotis daubentonii* (Kuhl, 1817) was first recorded in Greece in 1990 by Helversen and Weid [10]; cf. [93]. It appears to have a patchy distribution confined to the north of Greece. *M. daubentonii* has only been found in a few locations on the mainland and on the island of Lesvos in the region of North Aegean Sea (Figure 12). It is also rare in Albania [57], North Macedonia [76], the south of Bulgaria [77] and in Turkey [71]. The single observation of a few lactating females in a cave on the island of Lesvos in June 2017 may suggest seasonal migration from Turkey, but this hypothesis could only be confirmed with more systematic research. In the Prespa Lakes, northwest Greece, von Helversen and Weid [10] observed hundreds of Daubenton’s bats hunting over the water; it is, however, difficult to distinguish between *M. daubentonii* and *M. capaccinii* in their foraging grounds, therefore a proportion of these bats could have been the latter species (which occurs in sympatry). They also trapped several individuals of the species entering caves along the lakeshore; and some individuals were observed roosting in rock cavities and fissures in caves near the entrance (von Helversen, pers. comm.). Similar findings were confirmed in more recent years; nursery roosts were located in cavities near cave entrances, whilst small groups of males and non-reproductive females were roosting in crevices in rocky cliffs or overhangs along the lake shore [24]. No winter colonies are known.

*Myotis daubentonii* has been encountered from the lowlands (35 m a.s.l.) up to 1741 m a.s.l. The known roost elevation ranges from 250 m to 860 m a.s.l.

### 3.13. Myotis bechsteinii (Kuhl, 1817)

Beichstein’s bat *Myotis bechsteinii* (Kuhl, 1817) was first recorded in Greece in 1990 by von Helversen and Weid [10]. *M. bechsteinii* has a patchy distribution in mainland Greece and it has also been found on the island of Euboea in the Aegean Sea (Figure 13). Only one small nursery colony is known, in a disused bunker at 230 m.a.s.l from the late 1970s (early–mid August of 1976 and 1979) [10] but lactating females and juveniles have been recorded outside roosts in northern Greece. The earliest observation of a volant young has been made in the last week of July, suggesting it was born around late June–early July at the latest. Outside roosts, it was found mostly in woodland/forests with openings, pastures and semi-natural areas. The species has been encountered from the lowlands (20 m a.s.l.) up to 1175 m a.s.l. 

### 3.14. Myotis emarginatus (Geoffroy, 1806)

Geoffroy’s bat *Myotis emarginatus* (Geoffroy, 1806) was first recorded in Greece in 1881 by Winge [79]. It has been recorded in many locations in mainland Greece and on 18 islands of the Aegean and the Ionian Seas, the smallest being Ithaki (ca. 96 km^2^) in the Ionian Sea (Figure 14). Single individuals or colonies of variable sizes were found in 82 roosting sites including caves, and disused mines, concrete tunnels and buildings. The 16 largest colonies consisted of between 100 and 1060 individuals (a transient spring roost in a cavern of Central Macedonia). The largest nursery colonies (up to 600 females and juveniles) were found in both caves (including sea caves) and buildings.

Outside roosts, *M. emarginatus* was mainly encountered in woodland/forests, semi-natural areas and agricultural lands such as non-irrigated arable lands and olive groves, but mostly lands with significant areas of natural vegetation. The earliest record of lactating females comes from Crete and was in the first week of June. The species has been encountered from the sea level up to 1380 m a.s.l. The known roost elevation ranges from sea level to ca. 1000 m a.s.l. The main roosts of the species are surrounded mainly by sclerophyllous vegetation, mixed forest, sea and ocean, natural grassland and coniferous forest.

### 3.15. Myotis nattereri (Kuhl, 1817) 

Natterer’s bat *Myotis nattereri* (Kuhl, 1817) was first recorded in Greece in 1984 by Horáček and Hanák [101]. It is relatively common in mainland Greece, but it has only been recorded on four islands of the Aegean and the Ionian Seas, the smallest being Thassos (ca. 380 km^2^) in the northern Aegean Sea (Figure 15). *M. nattereri* is absent from Crete. Samos is the only island in the eastern Aegean Sea known to host the species, perhaps indicating the presence of a large colony on the island or in the Kusadasi area in western Turkey. Single individuals or colonies of variable sizes were found in 15 roosts, including caves, disused mines, concrete tunnels and buildings and bridges. The largest known colony of the species was recorded in a disused concrete drainage tunnel in the north of the Peloponnese, where up to 290 females were counted to form a nursery colony [44]. This is perhaps the largest known nursery colony of the species at all, since elsewhere it is known to form colonies no larger than 120 individuals, mostly in tree holes, bat boxes, rock and wall crevices and buildings [71]. All other known roosts in Greece hosted up to 15 individuals. In the Peloponnese, volant juveniles were caught in the first week of June. No winter colonies of the species are known yet.

Outside roosts, *M. nattereri* was found mostly in broadleaf woodland/forests and semi-natural areas. The species has been encountered from the sea level up to 1870 m a.s.l. (A. Alberdi, pers. comm.). The known roost elevation ranges from 5 m to 1215 m a.s.l. The largest roost of the species is surrounded mainly by transitional woodland shrub, sea, mixed forest and sclerophyllous vegetation.

### 3.16. Eptesicus anatolicus Felten, 1971

The presence of the Anatolian serotine bat *Eptesicus anatolicus* Felten, 1971 has been verified on five islands of the Region of the North Aegean and the prefecture Dodecanese, the smallest being Kastellorizo (ca. 12 km^2^) in the southeastern Aegean Sea (Figure 16). This is in line with its distribution along the Mediterranean coast of Turkey [78]. The species was first recorded in Rhodes in 1998 by von Helversen [102] as *E. bottae*, but it was later identified as *E. anatolicus* based on subsequent morphological and genetic analyses [87,103]. The species appears to be moderately common in Rhodes, but most records come from sites near the coast which are being deteriorated due to touristic development [104]; some of the localities described by von Helversen [102], have already been severely degraded because of this reason. Bats were captured on their way into three caves, in Rhodes and Samos, but no colonies are known to date. However, the capture of five pregnant females at two sites in Rhodes may indicate the presence of nursery colonies on the island. It has been recorded from sea level (5 m a.s.l.) up to 590 m a.s.l. usually at streams and estuaries. 

### 3.17. Eptesicus serotinus (Schreber, 1774)

The serotine bat *Eptesicus serotinus* (Schreber, 1774) was first recorded in Greece in 1855 by Lindermayer [5]. It is rather widespread in Greece, since it has been recorded in numerous locations on the mainland and on 10 islands of the Aegean and the Ionian Seas, the smallest being Samothraki (ca. 178 km^2^) in the northern Aegean Sea (Figure 17). Single individuals or usually small groups were found in 10 roosting sites, including caves, disused mines and other man-made structures. The only known large colony was located in a disused concrete drainage tunnel (95 m a.s.l.) in the region of West Greece, albeit with fluctuating numbers of individuals. Hundreds of *E. serotinus*, likely breeding females (many were pregnant), were recorded in June in the tunnel in two successive years. Over subsequent surveys in different months (May, July–September, in 2020) [44]; however, the numbers counted were much smaller (between 10 and 50). In July, a small nursery colony was observed (adult females and juveniles).

The earliest record of a volant young was in Crete, in the first week of July. Outside roosts, it was encountered mostly in woodland/forests and semi-natural areas with low vegetation. The species has been encountered from sea level up to 1490 m a.s.l., including both roosts and foraging/commuting sites. 

### 3.18. Nyctalus lasiopterus (Schreber, 1780)

The greater noctule bat *Nyctalus lasiopterus* (Schreber, 1780) was first recorded in Greece in 1964 by Wolf [105]. It is a relatively rare bat in Greece and has a patchy distribution on the mainland and has not been found on any of the islands (Figure 18). A tentantive recording of echolocation calls on the Greek side of the Prespa Lakes (northeast Greece) is not considered here, since its presence has not yet been confirmed by capture or direct observation of an individual [28]. Similarly, the species presence on the Albanian side of Prespa requires further investigation, since published records are based on echolocation calls only [106]. Only two roosts, both in trees, have been found, numbering a few individuals each; the exact size or function (maternity, hibernation or other) of these colonies was not specified ([105]; G. Tsougrakis, pers. Comm). Although von Helversen and Weid [10] suggested that only males spend the summer in Greece, females have also been encountered albeit less frequently (seven times), between early June and mid-September, in six localities of central and northern Greece. Males are much more abundant and widespread, and have been encountered 37 times, from late April to late September ([68]; Georgiakakis, unpublished data). Interestingly, females have not yet been found in any of the neighbouring Balkan countries, including Albania [57,106], North Macedonia [107] and Bulgaria [77,108], while only one female has been found in Turkey, where the species is rather rare [109]. Hibernation roosts are not known in Greece yet, therefore it is difficult to verify its presence in winter, similar to other tree roosting species.

*Nyctalus lasiopterus* has been encountered from the lowlands (40 m a.s.l.) up to 1190 m a.s.l.

### 3.19. Nyctalus leisleri (Kuhl, 1817)

Leisler’s bat *Nyctalus leisleri* (Kuhl, 1817) was first recorded in Greece in 1881 by Winge [79]. *N. leisleri* is common in mainland Greece and has also been encountered on 10 islands of the Aegean and the Ionian Seas, the smallest being Ikaria (ca. 255 km^2^) in the eastern Aegean Sea (Figure 19). Similar to other tree roosting species, there are, currently, no known roosts. In addition, its echolocation call parameters may overlap with those of *E. serotinus* and *Vespertilio murinus*; therefore, records considered here only include bats identified in the hand from captures at foraging/drinking and commuting sites. The vast majority of individuals caught were males, between April and October. Females were encountered much less frequently, mainly between August and October and less frequently in May (in total a few tens of individuals). Ιn Crete, only two male bats have been captured, in late July and in August. No juveniles have been recorded, suggesting that the species may not form nursery colonies in Greece in agreement with Helversen and Weid [10]. Furthermore, deaths of *N. leisleri* individuals at wind farms in the north east of Greece peaked in May, June and September, coinciding with the migratory season [35].

*Nyctalus leisleri* has been encountered from the sea level up to 2007 m a.s.l., mostly in forests and areas with low vegetation (natural or semi-natural). 

### 3.20. Nyctalus noctula (Schreber, 1774)

The noctule bat *Nyctalus noctula* (Schreber, 1774) is widespread in mainland Greece, albeit less common than Leisler’s bat (Figure 20). It was first recorded in 1855 by Lindermayer [5] on the island of Euboea, but since then, the only other island where it was captured was Thassos (ca. 380 km^2^) in the northern Aegean Sea ([9]; von Helversen unpublished data). Its echolocation calls have also likely been recorded on other islands, such as Kefalonia and Corfu in the Ionian Sea, but these records have not been accounted for here due to the partial overlap in its call parameters with other species (*N. leisleri*, *E. serotinus*, *V. murinus*). Its presence on these islands is yet to be confirmed through identification of bats in the hand, which is hindered by the difficulty in catching this elusive species. Females have been captured from late June to late September of different years mainly in the north of the country; in June 2004 a pregnant female was captured in northeast Greece, suggesting births in Greece. Juvenile bats were recorded in September 2002 roosting in a cavern in Central Macedonia together with adult females and males. The second known roost of the species in the country was located in a concrete road bridge in Thrace (northeast Greece): through captures in 2003 it was verified that it hosted male individuals only (May, June, September) and in 2004 counts on emergence confirmed the presence of 40 and 65 individuals on two separate occasions (August, October).

*Nyctalus noctula* has been encountered from the sea level (3 m a.s.l.) up to 1700 m a.s.l., mostly in woodland/forests, but also in variable cultivations and open areas with low vegetation.

### 3.21. Vespertilio murinus Linnaeus, 1758

The parti-coloured bat *Vespertilio murinus* Linnaeus, 1758 has a patchy distribution confined mainly to the north of mainland Greece (Figure 21). The species was first recorded in Greece in 1855 by Lindermayer [5] on the island of Euboea, in the Aegean Sea; to date, this remains the only island record. Its echolocation calls have also likely been recorded in more localities, but these records have not been accounted for due to the partial overlap in its call parameters with other species (*Nyctalus* spp. and *Eptesicus* spp.). *V. murinus* is also rare in Albania [57], North Macedonia [76,110] and Turkey [71], but more common in Bulgaria [77]. No roosts of the species are known in Greece. Twenty-three individuals have been identified in the hand overall, mainly captured over streams and ponds in forested areas. Of these, only three were females and all were caught in the autumn; the rest were males captured from mid-May to late October, suggesting that males remain in Greece in the summer, whilst females may only come to the country to mate and hibernate ([9]; von Helversen, pers. Comm). Similar observations from Bulgaria [77] and Albania [57] support this hypothesis.

*Vespertilio murinus* has been encountered from the lowlands (110 m a.s.l.) up to 1630 m a.s.l.

### 3.22. Hypsugo savii (Bonaparte, 1837)

Savi’s pipistrelle *Hypsugo savii* (Bonaparte, 1837) was first recorded in Greece in 1855 by Lindermayer [5]. It is one of the most common and widespread bat species in Greece, since it has been encountered in tens of locations on the mainland and on 25 islands of the Aegean and the Ionian Seas, the smallest being Thira (ca. 91 km^2^) in the southern Aegean Sea (Figure 22). Its echolocation calls partly overlap with those of *Pipistrellus kuhlii* and *P. nathusii*; a portion of sound recordings of doubtful origin may therefore belong to *H. savii*, which means it may be even more widespread and common than currently known. Small colonies or single individuals have been found in 13-day roosts, mostly buildings, bridges and rock fissures. Savi’s pipistrelles have further been captured entering or exiting 44 more sites (mainly caves and disused mines), but their type of use was unclear. The largest known colony (12 individuals) was recorded by Chaworth-Musters [111] in Agios Dionysios Monastery, Olympos Mts., Central Macedonia. The earliest record of volant young was in the first week of July, in ΝΕ Greece.

*Hypsugo savii* can be active throughout the year, particularly on the island of Crete [19], albeit less in winter. Outside roosts, it was recorded mostly in woodland/forests and open areas with low vegetation, but also cultivations of various types. In Crete, woodland and wetlands are preferred as feeding grounds [19]. The species has been encountered from the sea level up to 1950 m a.s.l. (A. Alberdi, pers. comm.). The known roost elevation ranges from sea level to 1500 m a.s.l. 

### 3.23. Pipistrellus kuhlii (Kuhl, 1817)

Kuhl’s pipistrelle *Pipistrellus kuhlii* (Kuhl, 1817) was first recorded in Greece in 1855 by Lindermayer [5]. It is likely the most common and widespread bat species in the country, since it has been recorded in several hundred locations on the mainland and on 27 islands of the Aegean and the Ionian Seas, the smallest being Kastellorizo (ca. 12 km^2^) in the southeastern Aegean Sea (Figure 23). It is often the most common species in urban environments [19,42]. Its echolocation calls largely overlap with those of *Pipistrellus nathusii* and partly with those of *H. savii* and, in the absence of trapping records, only the presence of its distinctive social calls can aid identification; many recordings of doubtful origin may therefore belong to *P. kuhlii*, which means it may be even more widespread than currently understood. Despite being extremely common, only 11 roosts have been found, in buildings and in fortification walls. Some summer and autumn colonies consisted of up to 53 individuals, all in various parts of buildings ([18,52]; own unpublished data), but most other colonies were substantially smaller.

The earliest recorded lactating females were caught from mid-June in Crete, whilst the first volant juveniles were captured from mid-July in Crete and central Greece, suggesting births start by the first week of June. No winter colonies are known yet. *P. kuhlii* can be active all year round, particularly in Crete [19], albeit less in winter. Outside roosts, it has been found mostly in cultivations of various types and semi-natural areas, urban environments and areas with low vegetation. In Crete, small wetlands and villages are preferred as feeding grounds [19]. The species has been encountered from sea level to medium elevation areas (0 to 1440 m a.s.l.) [20]. The known roost elevation ranges from 10 m to 800 m a.s.l.

### 3.24. Pipistrellus nathusii (von Keyserling et Blasius, 1839)

Nathusius’s pipistrelle *Pipistrellus nathusii* (von Keyserling et Blasius, 1839) was first recorded in Greece in 1978 by Pieper [112]. It is much less common than the morphologically very similar *P. kuhlii* in Greece, since it appears to have a patchy distribution on the mainland and it has mostly been encountered in the north of the country (Figure 24). Its echolocation calls, however, largely overlap with those of *P. kuhlii* and partly those of *H. savii*. Therefore, unless its distinctive social calls are present in sound recordings or bats are identified in the hand, recordings cannot be safely attributed to either species, which means it may be present in more areas than currently known. *P. nathusii* is still, however, much less common in trapping records than *P. kuhlii.* A few bats have also been identified in the hand on three islands of the Aegean and the Ionian Seas, the smallest being Kerkyra (Corfu, ca. 593 km^2^) in the Ionian Sea. Small colonies or single individuals have been encountered in 12 roosting locations, including fissures in caverns, rock overhangs, rock crevices, disused buildings, bridges, trees and bat boxes. The largest known colony was found in a cavern in Central Macedonia consisting of up to 40 individuals in spring and autumn across different years; it was likely of transient nature, because it was absent in the winter and summer. Similarly, a smaller colony of up to 22 individuals was present in an abandoned building in the same prefecture in late spring and autumn of different years. The trapping of a pregnant female in northern Greece in June 1989 [9] denotes that the species reproduces in Greece. Females, however, have otherwise rarely been recorded between May and September, and only one juvenile has been observed, in northeastern Greece [17]. The vast majority of individuals captured were adult males. In Crete, the species has only been encountered twice (a captured male and a recording of social calls), in May 2003, suggesting seasonal migration. Deaths at wind farms in northeastern Greece peaked in April and September, coinciding with the migration period, but also in June [35].

Outside roosts, *P. nathusii* was found in a variety of habitat types, from urban environments and cultivated areas to woodland/forests and wetlands. The species has been encountered from the sea level up to 1270 m a.s.l. The known roost elevation ranges from 5 m to 640 m a.s.l.

### 3.25. Pipistrellus pipistrellus (Schreber, 1774)

The common pipistrelle *Pipistrellus pipistrellus* (Schreber, 1774) was first recorded in Greece in 1964 by Laar and Daan [113]. It is one of the most common bat species in Greece. It has been recorded in several hundred locations on the mainland and on 22 islands of the Aegean and the Ionian Seas, the smallest being Kastellorizo (ca. 12 km^2^) in the southeastern Aegean Sea (Figure 25). *P. pipistrellus* is absent from Crete, where it is replaced by Hanák’s pipistrelle. It has been found in eight roosting sites, all buildings. At least two of these hosted nursery colonies; one observed between the external wall and the sign of a kindergarten including ca. 45 individuals; the other observed within the wall of a remote pumping station in Rhodes including ca. 20 individuals. No winter roosts are known. In Rhodes, a foraging juvenile was trapped in the second week of June 2014; this is the earliest known record of a flying juvenile pipistrelle in Greece. The species may be active all year round, albeit less in winter.

Outside roosts, it was found in a wide range of habitats, mainly woodland/forests, cultivated areas and urban environments. Common pipistrelles have been encountered from the sea level up to 1870 m a.s.l. (A. Alberdi, pers comm.). The known roost elevation ranges from sea level to 1200 m a.s.l.

### 3.26. Pipistrellus pygmaeus (Leach, 1825)

The soprano pipistrelle *Pipistrellus pygmaeus* (Leach, 1825) was first recorded in Greece in 1964 by Laar and Daan ([113], as *P. pipistrellus*); first confirmed records of *P. pygmaeus* were published simultaneously by Hanák et al. [9] and Mayer and von Helversen [114]. It is very common in mainland Greece, albeit less common than the morphologically very similar *P. pipistrellus* (Figure 26). The species has been recorded in several hundred locations on the mainland; it is also present on at least seven islands of the Aegean and the Ionian Seas, the smallest being Kos (ca. 290 km^2^) in the eastern Aegean Sea. *P. pygmaeus* is absent from Crete and the semi-arid islands of South Aegean (Cyclades and most of the Dodecanese). Very few roosts of the species have been found, in a variety of roost types. In the autumn of 2015, two mating groups consisting of one male and two to three females were found in bat boxes in the Peloponnese (unfortunately all bat boxes were removed in the following years). In the autumn of 2013, a roost was located in a rock cavity at the entrance of a vertical cave on the island of Kefalonia. The largest roost of the species currently known, was found in an abandoned hotel, which is part of the tourist resort in Kaiafas Lake on the west of the Peloponnese (in the region of West Greece). The roost hosted a large nursery colony of 350 individuals in 1995 (von Helversen, unpublished data); it was still present in 2018 [41]. Unfortunately, despite its importance, the roost is not protected in practice, meaning that there are no measures in place if the building is to be renovated or demolished in future.

The earliest records of volant young come from the mid-late June, on Kefalonia island. Outside roosts, it was encountered mostly in woodland/forests and cultivated land, often close to or over water surfaces. The species has been encountered from the sea level up to 1700 m a.s.l. 

### 3.27. Pipistrellus hanaki Hulva et Benda, 2004

Hanák’s pipistrelle *Pipistrellus hanaki* Hulva et Benda, 2004, is a recently described species, present only on the island of Crete (Figure 26) and in Cyrenaica, Libya [115,116,117]. It was first recorded in Greece in 1959 by Kahmann ([118], det. as *P. pipistrellus*). The species appears to prefer broadleaf forests and old tree cultivations as foraging grounds, but uses a wide variety of roost types: trees of several species (e.g., oaks, olive trees, carob trees), rock crevices, buildings and electricity posts [22]. There are 35 known roosts and the largest among these (a tile roof in a house) hosted 17 individuals (post-lactating females and possibly juveniles). Only two nursery colonies are known, both in trees (2–15 females and juveniles in each). Juveniles in flight were captured as early as in the first days of July [22]. No winter roosts are known. *P. hanaki* It is active all year round, albeit less in winter [19].

The species has been encountered from the lowlands (5 m a.s.l.) up to 1520 m a.s.l. The known roost elevation ranges from 380 m to 1000 m a.s.l.

### 3.28. Barbastella barbastellus (Schreber, 1774)

The western barbastelle *Barbastella barbastellus* (Schreber, 1774) was first recorded in Greece in 1987 by Volleth [93]. It appears to be widespread in mainland Greece, although it has a patchy distribution (Figure 16) and has been located in a moderate number of locations including only one island, Thassos in the northern Aegean Sea. Εcholocation calls of the species have presumably been recorded in Euboea, Corfu and the Prespa Lakes (northwest Greece) [28], but these records are considered doubtful, since they fall outside the verified species range. This approach is further supported by the absence of the species from eastern Central Greece and its rather scarce distribution in south Albania [57], North Macedonia [76], the region of West Macedonia and the lowlands of the region of Epirus in Greece. Small numbers of *B. barbastellus*, mostly single males, were netted at the entrance of four underground sites (caves and disused mines), in the summer and autumn across several years, but the use of these sites is unknown; to date, no colonies have been located in the country. In northern Greece, pregnant females were captured by late June, indicating that the species reproduces in the country. Only two juveniles were captured so far, in September 2002, in Central Greece.

*Barbastella barbastellus* has been encountered from the lowlands (81 m a.s.l.) up to 1550 m a.s.l., and outside roosts mostly in heterogeneous agricultural and forested areas. In the Red List of Threatened Species of Greece [2] it is classified as Endangered.

### 3.29. Plecotus auritus (Linnaeus, 1758) 

The brown long-eared bat *Plecotus auritus* (Linnaeus, 1758) was first recorded in Greece in 1881 by Winge [79]. It may be one of the rarest bat species in Greece; it is likely that this is the southern limit of its European distribution (Figure 27). Similarly, it is rare in Albania [57] and North Macedonia [90], but more common in Bulgaria [77] and Turkey [71]. These differences may be attributed to variation in sampling effort and the difficulties in hand identification of the species. The vast majority of individuals were found in forested areas in the north of the mainland. Older records published under this name were later attributed to *P. kolombatovici* [9] and references therein. A small colony in a small cave in central Greece found in August 1981 and 1988 was attributed to this species [10] but these records may also belong to *P. kolombatovici* (von Helversen, pers. comm.).

*Plecotus auritus* has been encountered from the lowlands (10 m a.s.l.) up to 1740 m a.s.l.

### 3.30. Plecotus macrobullaris Kuzâkin, 1965

The Alpine long-eared bat *Plecotus macrobullaris* Kuzâkin, 1965 was first recorded in Greece in 2001 by Hanák et al. [9]. It has only been recorded in a few localities in the regions of West Macedonia (Prespa Lakes), Epirus, Central Greece (Pindos Μountains) and on the island of Crete (Figure 27). *P. macrobullaris* clearly appears to follow its distribution from the Pyrenees to the Alps, along the Dinaric Μountains to Pindos Μountain range. Only one record is known from North Macedonia [119], and it may be absent from Bulgaria [71,77]. Interestingly, it has not yet been found in the Peloponnese, which lies between Pindos and Crete. A sub-adult female was roosting during daytime in a building in the south of Crete in late July 2011. In Crete, single individuals have also been caught in late spring and autumn at the entrance of five underground sites (four caves and a disused mine), but the use of these sites was not clear.

*Plecotus macrobullaris* has been encountered from the lowlands (46 m a.s.l.) up to 1895 m a.s.l. (A. Alberdi, pers comm.).

### 3.31. Plecotus austriacus (Fischer, 1829) 

The presence of the grey long-eared bat *Plecotus austriacus* (Fischer, 1829) has been confirmed in 21 localities in the north of Greece (regions of West Macedonia, Central Macedonia, East Macedonia and Thrace, Thessaly). Additionally, there is one record from Fokida in Central Greece [120] and one from Limnos Island (ca. 478 km^2^) in the northeastern Aegean Sea (Figure 28); however, the latter records require further investigation, since they are rather old and far from all other records. The species was first recorded in Greece by Kock [121,122]. Solitary individuals or small groups have been found in 14 roosts, including caves, disused mines, bridges and underground military shelters, in the autumn. Some of these sites were used as day roosts and others as night roosts. No nursery or winter roosts are known.

The species has been encountered from the lowlands (40 m a.s.l.) up to 1120 m a.s.l., including both roosts and foraging/commuting areas. 

### 3.32. Plecotus kolombatovici Đulić, 1980

The Mediterranean long-eared bat *Plecotus kolombatovici* Đulić, 1980 is widespread in mainland Greece and it is also found on 13 islands of the Aegean and the Ionian Seas, the smallest being Kalymnos (ca. 110 km^2^) in the southeastern Aegean Sea (Figure 28). It appears to be absent from the north of the mainland, excluding Chalkidiki [56], where *P*. *auritus* and *P*. *austriacus* are present. Interestingly, it has been found in several sites in Albania, including the borders with Greece [57], but not in North Macedonia or Bulgaria [71]. The first Greek record of the species was in Crete in 1967 by Martens [123] although it was identified as *P. austriacus*. *P. kolombatovici* has been found in 22 roosting sites, including caves, disused mines and bunkers, a railway tunnel and buildings. The largest known colony was found in a railway tunnel in north Peloponnese (Region of Western Greece); it was a maternity colony consisting of up to 120 individuals from May to October. No winter colonies are known.

Outside roosts, *P. kolombatovici* was found mostly in forested areas and close to freshwater. The species has been encountered from the lowlands (5 m a.s.l.) up to 1490 m a.s.l., including both roosts and foraging/commuting areas. 

### 3.33. Miniopterus schreibersii (Kuhl, 1817)

Schreiber’s bat *Miniopterus schreibersii* (Kuhl, 1817) was first recorded in Greece in 1954 by Pohle [124]. It is one of the most widespread and abundant species in Greece, since it has been found in tens of locations in mainland Greece and on 16 islands of the Aegean and the Ionian Seas, the smallest being Ikaria (ca. 255 km^2^) in the eastern Aegean Sea (Figure 29). Single individuals or colonies of variable sizes have been found roosting in 139 underground sites, including caves, disused mines and concrete drainage tunnels. The largest known colony was located in the Cave of the Lakes, in the region of West Greece, in winter; it is, in fact, the largest known bat colony in Greece, with fluctuating numbers reaching up to 25,000 individuals [43]. The fluctuation in numbers indicates that the species may also be active in winter, on warm and calm nights, as it has also been shown elsewhere in Greece. Forty-six other colonies exceeded 100 individuals each, with nursery colonies consisting of up to 6000 individuals; all nursery colonies were found in caves (including sea caves) and disused mines. Similar to *Myotis capaccinii*, *M. schreibersii* may use a network of roosting sites within and between seasons. For example, in Thrace (NE Greece), small mines were used by highly fluctuating numbers of individuals, reaching over 2000 bats on some occasions in spring; in summer and winter, these mines were usually used by less than 100 bats, indicating that the species mostly used them as transient roosts. Several thousand Schreiber’s bats roosted in a cave along the shore of Prespa Lake (NW Greece) in summer, but only a few hundred were present within the cave in the autumn [24].

Outside roosts, *M. schreibersii* has been encountered mostly in woodland/forests, agricultural areas and open natural areas with low vegetation. Juveniles in flight have been captured as early as the first week of July, in northeastern Greece. The species has been encountered from the sea level up to 1500 m a.s.l. The known roost elevation ranges from sea caves to 1400 m a.s.l. The main roosts of the species are surrounded mainly by sclerophyllous vegetation, mixed forest, sea, non-irrigated arable land and land principally occupied by agriculture, with significant areas of natural vegetation.

### 3.34. Tadarida teniotis (Rafinesque, 1814)

The European free-tailed bat *Tadarida teniotis* (Rafinesque, 1814) was first recorded in Greece in 1855 by Lindermayer [5]. It appears to be common in Greece, since it has been located in a few hundred locations on the mainland and on 27 islands of the Aegean and the Ionian Seas, the smallest being the uninhabited islet of Dragonada (ca. 3 km^2^), northeast of Crete (Figure 30). Twelve roosts have been located, including in buildings, bridges and fissures in rocky cliffs, overhangs and above cave entrances. Colonies in these roosts consisted of up to 30 individuals, including two nursery colonies in rock fissures (on the islands of Crete and Limnos), but some hosted only single individuals. No winter roosts are known. The species is active all year round, particularly in Crete [19], albeit less in winter. In the autumn of 2002, over 100 individuals were trapped in a very small balcony in a large city in the north of Greece; however, the location of the roost or roosts of origin remains unknown [32].

Outside roosts, *T. teniotis* it was recorded flying over a wide variety of habitats mostly including woodland/forests, agricultural areas, open natural areas with low vegetation and urban environments. In Crete, wetlands, shrublands and villages are preferred as feeding grounds [19]. The species has been encountered from the sea level (0 m a.s.l.) up to 1900 m a.s.l. The known roost elevation ranges from 0 sea level to 890 m a.s.l.

### 3.35. Rousettus aegyptiacus (Geoffroy, 1810)

The Egyptian fruit bat *Rousettus aegyptiacus* (Geoffroy, 1810) was only recently (in 2018) recorded in Greece, on one occasion on the island of Kastellorizo in the southeast of the Aegean Sea (Figure 20), ca. 3 km from the south-Anatolian coast [50]. At least three individuals were observed foraging in a mulberry tree, in the capital of the island. A second expedition failed to verify its presence. We can, therefore, assume that it occasionally visits Kastellorizo from the coast of Turkey to forage and no permanent population has been established on the island until further research confirms the opposite. Our hypothesis is supported by the documentation of nightly movements of the species over several tens of kilometres in Israel [125] and Cyprus [126].

## 4. Discussion

In this article we review, update and summarize available information on the distribution and ecology (mainly in terms of foraging and roosting habitat use) of bats in Greece. To date, the occurrence of 35 bat species is well documented in the country (Table 1). Although *M. mystacinus* was thought to be widespread in Greece, currently its presence in the country remains unconfirmed in the light of recent molecular genetic studies, but is possible in northeastern Greece ([9,96]; F. Mayer, pers. Comm). The tentative recording of echolocation calls of *E. nilssonii* in the area of Prespa Lakes (northwestern Greece) is not considered here, because it was not confirmed by capture or direct observation of an individual [28].

In Greece, the most common species appear to be *P. pipistrellus*, *H. savii*, *R. ferrumequinum*, *P. kuhlii* and *T. teniotis* with over 500 occurrence records each. *E. anatolicus*, *P. auritus*, *P. macrobullaris*, *M. alcathoe*, *M. brandtii* and *R. aegyptiacus* are the rarest bats, with less than 20 occurrence records each. Among cave-roosting species, *M. schreibersii* and *M. blythii* appear to be the most abundant, with several large colonies recorded; *R. mehelyi* appears to be the least abundant.

The highest number of species was observed in northern and central mainland Greece (Table 1). This may be explained by a higher variation in habitat types and the proximity to other countries of the Balkan peninsula, therefore hosting species of more northern origin (e.g., *Myotis alcathoe*, *M. brandtii*, *M. daubentonii*, *Plecotus auritus*, *P. austriacus* and *Vespertilio murinus*). Species richness, however, may also reflect variation in sampling effort. The relatively small number of species in the prefecture of Attica, Central Greece (Table 1), where Athens, the capital of Greece, is located may be explained by the high degree of urbanisation relative to its small size (hosting ca. 50% of the population of Greece) and the limited bat research conducted in that area.

### 4.1. Bats οn Greek Islands

Throughout all the years of surveys in the country, overall, 31 bat species were found on at least 1 Greek island, with 14 species recorded on more than 10 islands of both the Aegean and the Ionian Seas (Table 2). The highest number of species was documented in Euboea (20 species), followed by Corfu and Lesvos (19 species), Kefalonia and Samos (18 species) and Crete (17 species). These are among the largest islands of Greece and the number of species may reflect their size, but also their proximity to the mainland, variation in habitat types and sampling effort. Many smaller islands in Greece have not or barely been surveyed for bats. Crete is the largest island in the country and has been studied extensively; it hosts, however, a lower number of species compared to smaller islands, presumably due to its geographic isolation. Some species of Asian affiliation (*Eptesicus anatolicus* and *Rousettus aegyptiacus*) only occur on East Aegean islands close to Turkey. Presumably these islands constitute the western limit of the distribution of *E. anatolicus*, whilst *R. aegyptiacus* is known to occur in Anatolian sites situated west of Kastellorizo (see [127]). *Pipistrellus hanaki* only occurs in Crete, which is likely the northern limit of its distribution. More systematic surveys and research are necessary to better understand the distribution, ecology and conservation needs of bats on the Greek islands.

Many Greek islands constitute popular tourist attractions, welcoming an increasing number of visitors each summer. Ongoing changes in land use in order to facilitate mass tourism has led to the loss of a significant part of natural habitats, such as woodlands and wetlands [104] and the abandonment of traditional cultivations. The low bat species richness and small size of colonies observed in some of the most heavily visited islands may be the result of such pressures, denoting the need for touristic activities more friendly towards the natural environment.

### 4.2. Altitudinal Distribution

The number of species appears to decrease with elevation: currently, 32 bat species are known to roost or forage/commute above 1000 m a.s.l., 21 species above 1500 m a.s.l. and only two species have been encountered above 2000 m a.s.l. (Table 3). Activity also decreases with elevation, depending on species and habitat type ([20,128]; own unpublished data). To date, no nurseries of any species have been found above 1000 m a.s.l., but this can be expected in some species, e.g., *Plecotus macrobullaris*. Daily torpor has been observed in most cave–dwelling species, but true hibernation has been documented only in *Myotis blythii*, above 1000 m a.s.l., on the islands of Crete and Kefalonia. More surveys at higher elevation areas are required to understand the altitudinal distribution and activity of bat species in Greece, particularly given that much of the country’s surface is mountainous.

### 4.3. Sea and Lake Caves

To date, 17 caves along the Greek coast and lake shores have been documented to provide roosting space to six bat species overall (*Myotis blythii*, *M. emarginatus*, *Miniopterus schreibersii*, *Rhinolophus ferrumequinum*, *R. euryale* and *R. hipposideros*), but likely more such caves are present. Some of these caves host nursery colonies numbering up to a few thousand females and pups, and may only be used in summer. For example, a sea cave on the island of Kefalonia surveyed in the autumn had no bats present but a large amount of guano indicated its use from many bats earlier in the year. The extensive and largely inaccessible coastline may provide roosting space to many more bats and deserves systematic surveys. Similarly, caves (some semi-flooded) along the shore of the lakes of Prespa, Kastoria and Volvi hosted significant numbers of various bat species; these also require further research and appropriate conservation action.

### 4.4. Conservation and Research Priorities

Knowledge on the distribution and ecology of bats in Greece and systematic surveys currently remain limited for various reasons including lack of funding, although it is certainly more advanced compared to previous decades. Many records of bat species in Greece come from relatively older surveys of roosts or foraging/commuting sites, hence they are likely outdated. The land use change has been dramatic in Greece over the past decades but as surveys and active protection of bat populations remain insufficient, many important sites may have been lost since they were last surveyed. For example, the galleries in Kimmeria (Xanthi, East Macedonia and Thrace) used to host several hundreds of bats from nine species [9,98,122,129]; in 2014 the entrance of most of these galleries was completely buried without any consideration for bats, and an unsuitable grill was placed at the entrance of the last open gallery, preventing free access to bats. Removing the grill was followed by a partial recovery of the colony, but long-term management and conservation actions are still missing. Management and conservation of roosts in artificial sites is absent, despite the high legal protection status of all bat species (Table 1). This issue needs to be addressed and the protection of artificial roosts prioritised, particularly as increasingly more roosts are discovered in buildings and underground sites such as mines [130,131].

In more recent years, the implementation of conservation and management programs run by the Natural Environment and Climate Change Agency (N.E.C.C.A.) and other research and conservation programs, such as EUROBATS Project Initiative [22] and LIFE GRECABAT (LIFE17 NAT/GR/000522) [69], are helping to fill knowledge gaps and to improve the conservation status of bats in Greece. Within the framework of the Life-Nature project LIFE GRECABAT, an action plan was recently produced for species roosting in underground sites, the first of its kind in Greece. The action plan offers an overview of the knowledge on cave bats in the country and describes appropriate conservation actions to be implemented at a national level [131], but it remains yet to be implemented. Caving clubs and independent cavers have been greatly contributing to this direction, communicating and helping to discover new bat roosts; this is a citizen science activity that merits institutional support.

Τhe National Database of the Chiroptera of Greece (NDCG), National History Museum, University of Crete, includes data on more than 1000 bat roosts, including both underground and overground sites (natural or artificial), hosting from single male individuals to large winter or maternity colonies from several species. There are, however, many more sites that remain undetected, particularly caves, given the rough terrain of the country such as hardly accessible or inaccessible mountain slopes and rocky sea shores. Very few hibernacula are known, and roosts of tree roosting bats are barely known; knowledge on roosting sites in man-made structures also remains very limited. Commuting sites and foraging areas in particular on mountains are little known and their use is poorly understood. The rapid licensing and installation of large-scale renewable energy facilities, including both wind farms and solar panels, along many of the mountain ranges and other natural areas in Greece, with no or limited consideration for bats in the Environmental Impact Assessments (EIAs), may therefore lead to the destruction of important sites before they are even discovered [132]. The same applies to dramatic changes in land use for development and intensive agricultural practices. It is imperative that surveys are implemented to locate, study and appropriately protect potentially important sites used by bats, whether for roosting, foraging or commuting.

Overall, the use of currently known roosting sites by bats in Greece is generally poorly understood, with a few exceptions, e.g., [14,16,43,44,69,133,134,135]. The same applies to foraging and commuting sites. Systematic and seasonal surveys are necessary to better understand the roosting and foraging ecology of bats in Greece, including hibernation, nursery and swarming/mating seasons.

Transboundary collaboration across countries is recommended for the study and conservation of bats near the borders. In Thrace (northeastern Greece), collaboration with Bulgarian and Turkish researchers showed that bats cross the borders on a nightly [13] or seasonal basis, explaining seasonal fluctuations in numbers of roosting populations [14,16]. Similarly, transboundary collaboration in the area of Prespa Lakes (northwestern Greece), resulted in a conservation action plan for the bats of Prespa [24] which would have been incomplete without the input of scientists from the neighbouring countries.

## 5. Conclusions

Despite its small size, Greece hosts one of the most diverse bat faunas in Europe with a wide range of roosting and foraging habitat requirements. The annual use of the majority of known roosting sites, however, remains poorly understood, with a few exceptions, e.g., [14,16,43,44,69,133,134,135] and many roosts are unknown. The use of foraging and commuting habitats by bats is also poorly understood for the majority of species and areas of Greece. Systematic seasonal surveys are therefore necessary and funding should be provided to better understand the roosting and foraging ecology of bats in the country (including hibernation, nursery and swarming/mating seasons), and impact of development projects on bats should be included in EIAs. This is imperative given their legal protection status on a national and international level, and the pressures on their habitats resulting from dramatic land use changes for development purposes and large-scale renewable energy installations.

## Figures and Tables

**Figure 1 animals-13-02529-f001:**
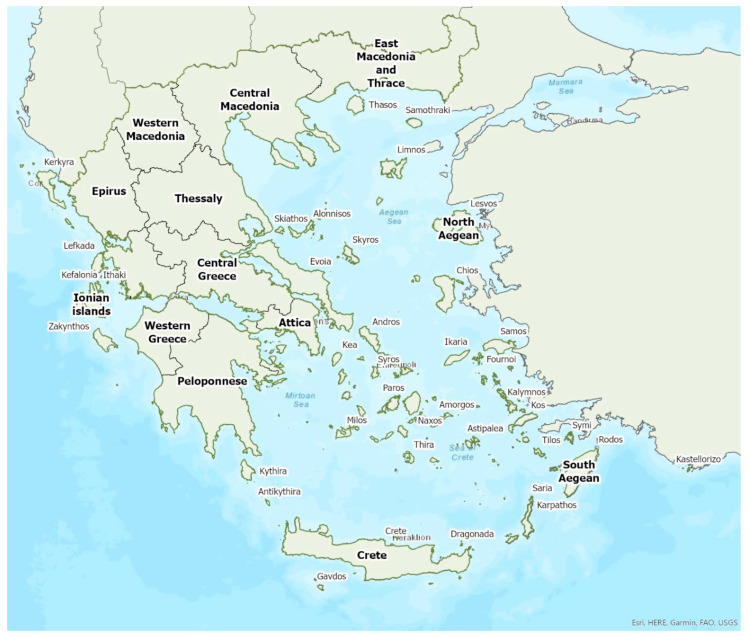
Map of Greece with the names of Administrative Regions and the islands surveyed for bats.

**Figure 2 animals-13-02529-f002:**
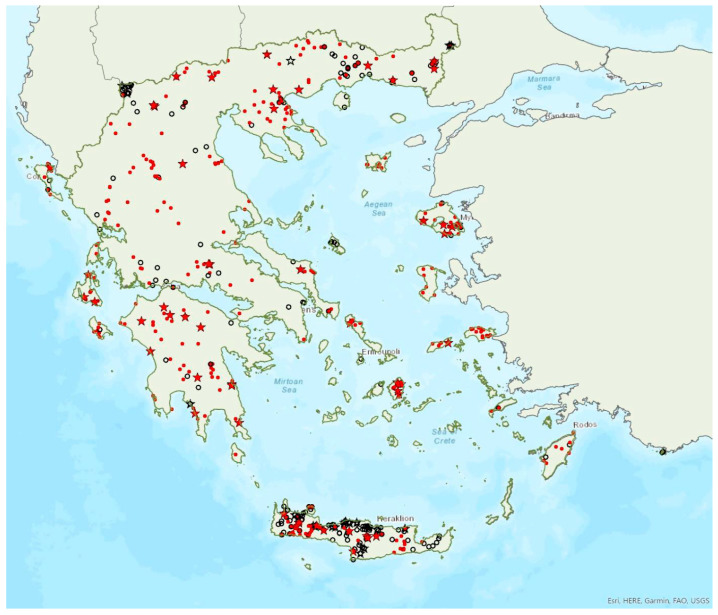
Records of *Rhinolophus ferrumequinum* (Schreber, 1774) in Greece. Empty black symbols: published records. Filled red symbols: unpublished records. Stars: main roosts. Circles: minor roosts and foraging-commuting sites.

**Figure 3 animals-13-02529-f003:**
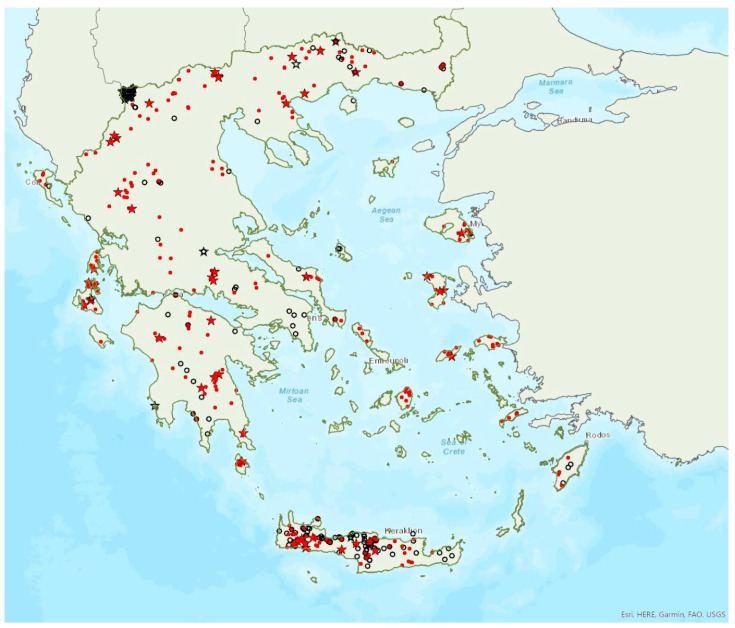
Records of *Rhinolophus hipposideros* (André, 1797) in Greece. Empty black symbols: published records. Filled red symbols: unpublished records. Stars: main roosts. Circles: minor roosts and foraging-commuting sites.

**Figure 4 animals-13-02529-f004:**
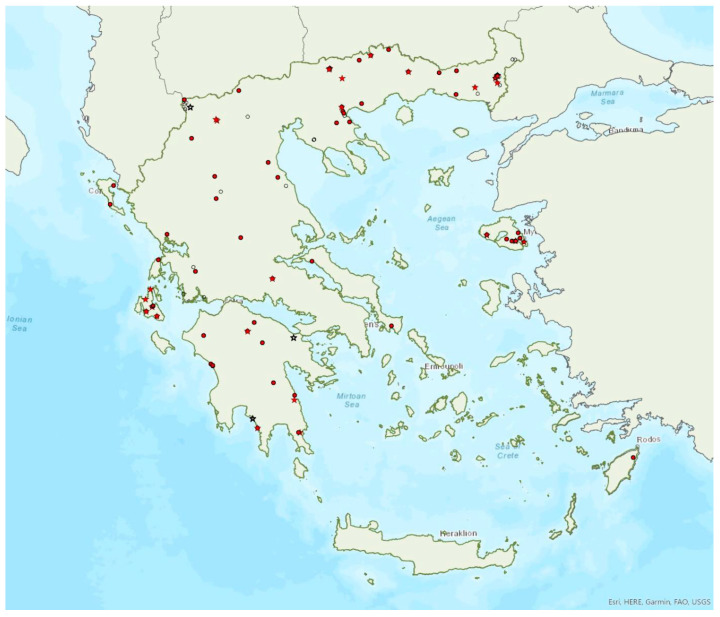
Records of *Rhinolophus euryale* Blasius, 1853 in Greece. Empty black symbols: published records. Filled red symbols: unpublished records. Stars: main roosts. Circles: minor roosts and foraging-commuting sites.

**Figure 5 animals-13-02529-f005:**
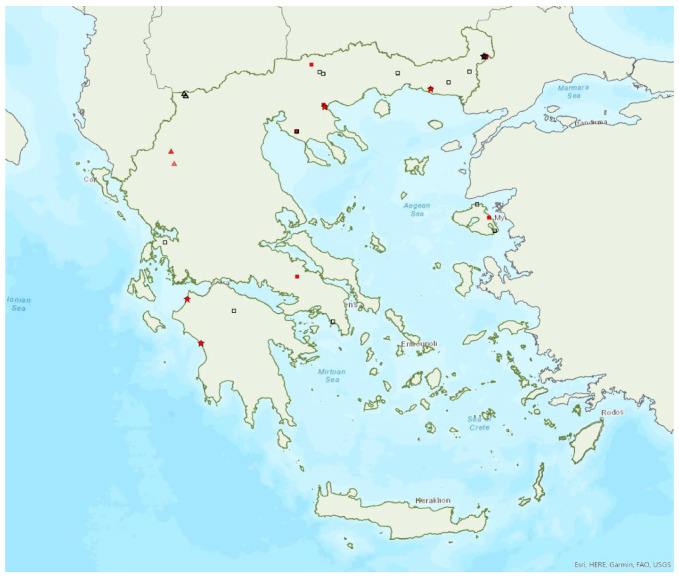
Records *Rhinolophus mehelyi* Matschie, 1901 (empty black squares: published records; filled red squares: unpublished records; empty black stars: published main roosts; filled red stars: unpublished main roosts) and *Myotis brandtii* (Eversmann, 1845) (empty black triangles: published records; filled red triangles: unpublished records) in Greece.

**Figure 6 animals-13-02529-f006:**
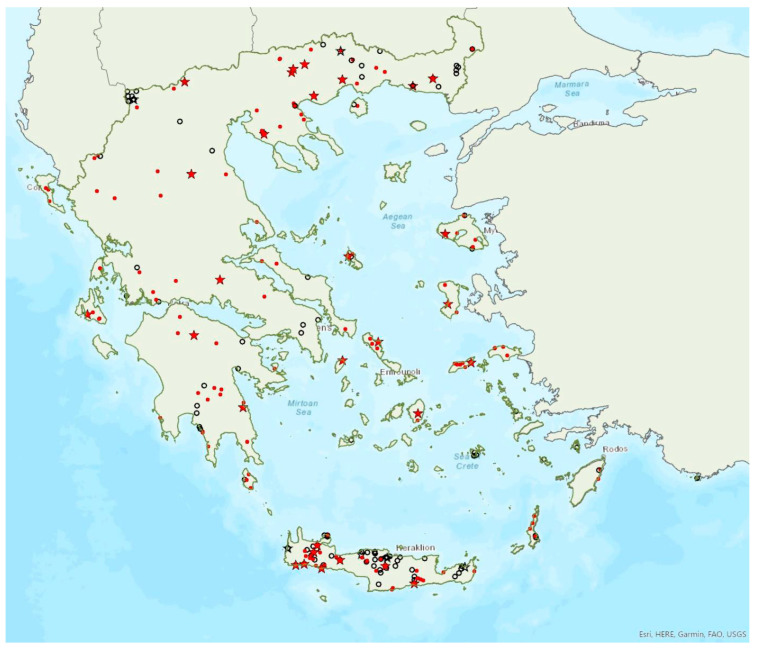
Records of *Rhinolophus blasii* Peters, 1866 in Greece. Empty black symbols: published records. Filled red symbols: unpublished records. Stars: main roosts. Circles: minor roosts and foraging-commuting sites.

**Figure 7 animals-13-02529-f007:**
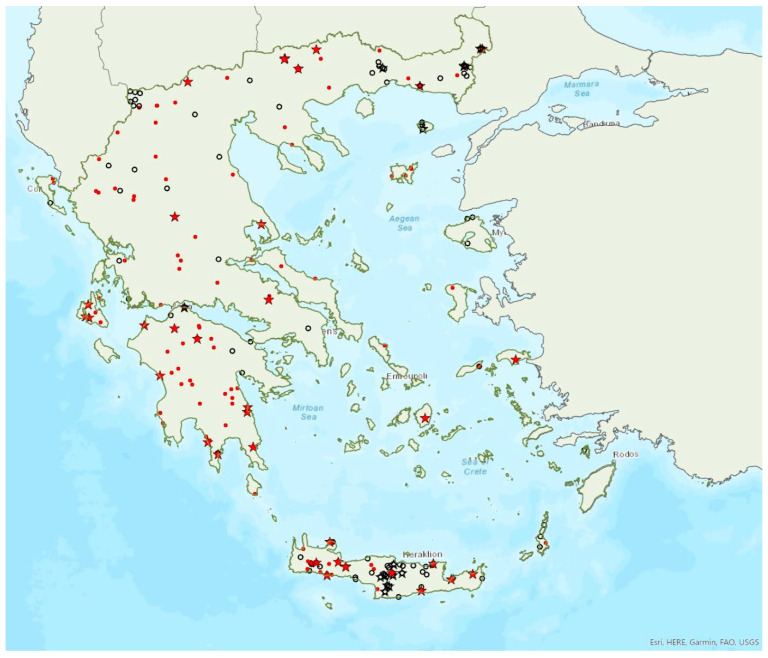
Records of *Myotis blythii* (Tomes, 1857) in Greece. Empty black symbols: published records. Filled red symbols: unpublished records. Stars: main roosts. Circles: minor roosts and foraging-commuting sites.

**Figure 8 animals-13-02529-f008:**
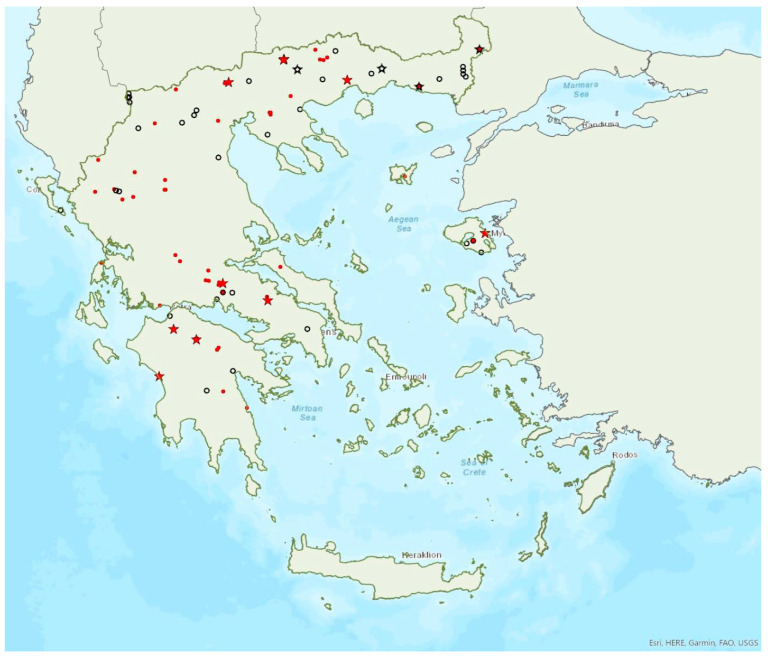
Records of *Myotis myotis* (Borkhausen, 1797) in Greece. Empty black symbols: published records. Filled red symbols: unpublished records. Stars: main roosts. Circles: minor roosts and foraging-commuting sites.

**Figure 9 animals-13-02529-f009:**
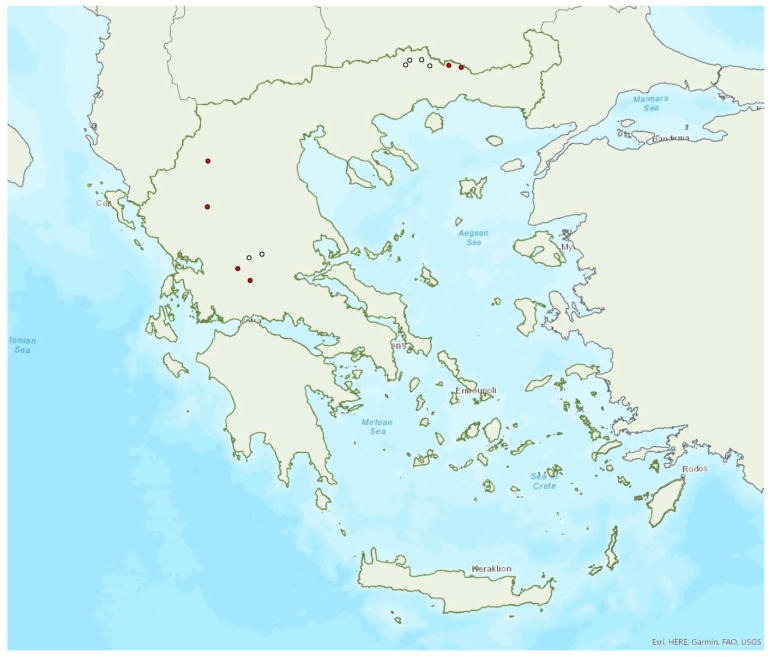
Records of *Myotis alcathoe* von Helversen and Heller, 2001 in Greece. Empty black symbols: published records; filled red symbols: unpublished records.

**Figure 10 animals-13-02529-f010:**
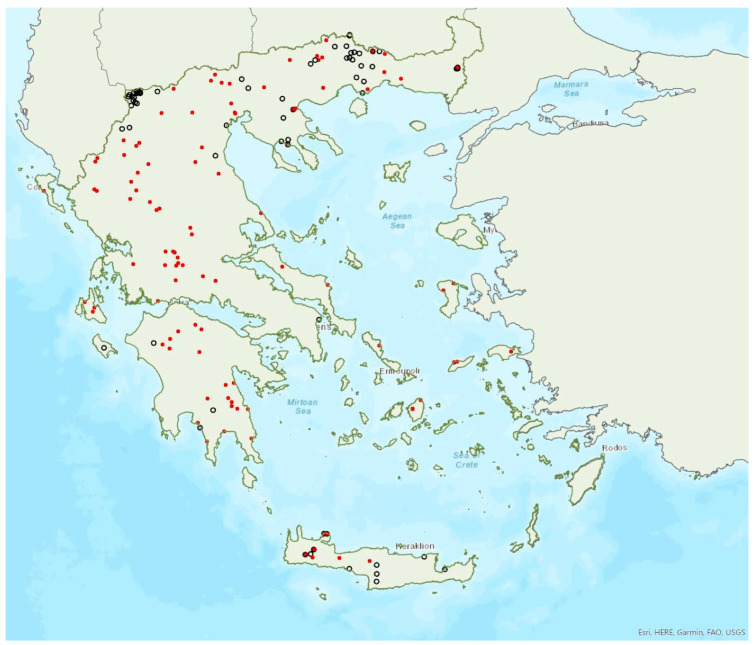
Records of *Myotis davidii* (Peters, 1869) in Greece. Empty black symbols: published records; filled red symbols: unpublished records.

**Figure 11 animals-13-02529-f011:**
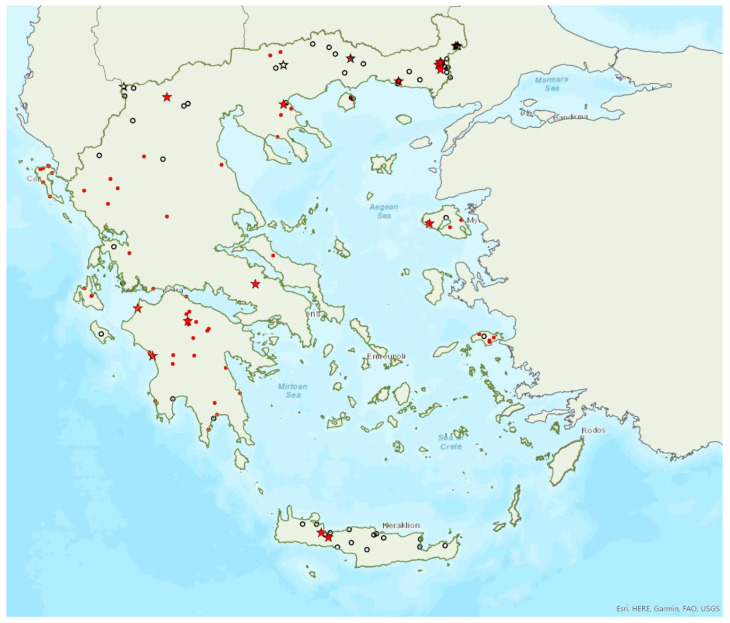
Records of *Myotis capaccinii* (Bonaparte, 1837) in Greece. Empty black symbols: published records. Filled red symbols: unpublished records. Stars: main roosts. Circles: minor roosts and foraging-commuting sites.

**Figure 12 animals-13-02529-f012:**
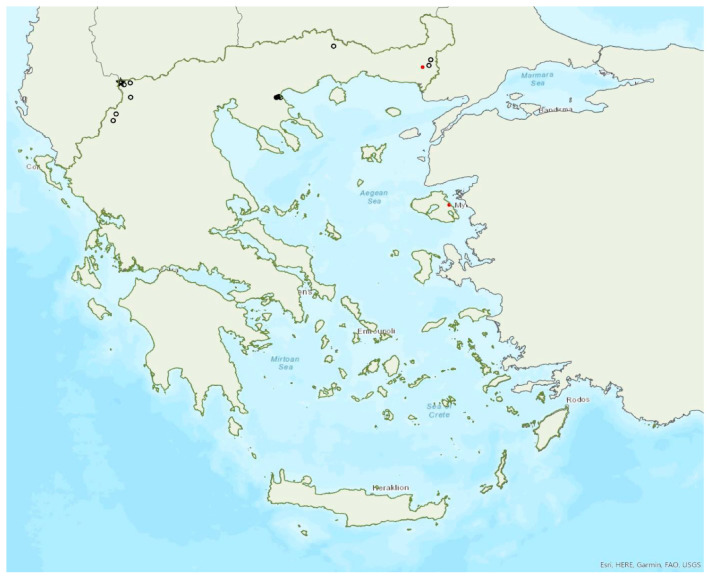
Records of *Myotis daubentonii* (Kuhl, 1817) in Greece. Empty black symbols: published records. Filled red symbols: unpublished records. Stars: main roosts. Circles: minor roosts and foraging-commuting sites.

**Figure 13 animals-13-02529-f013:**
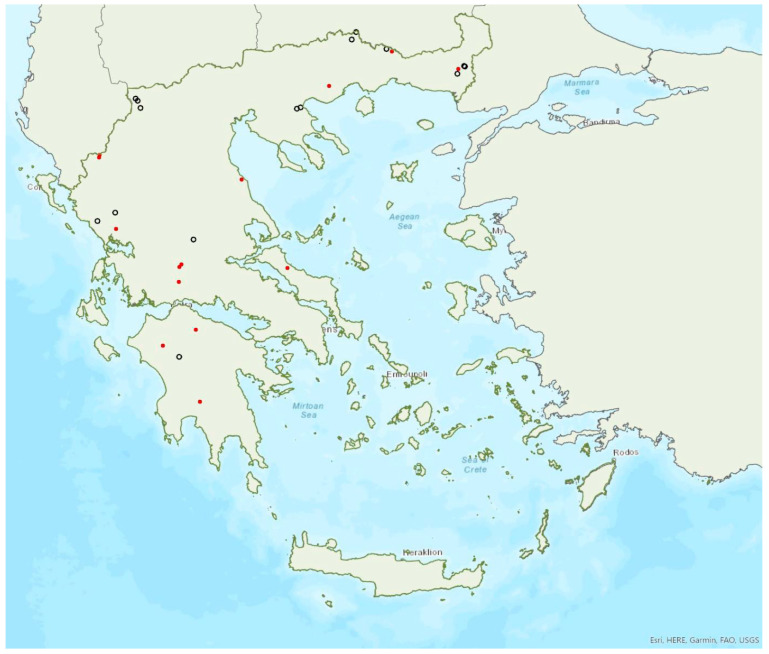
Records of *Myotis bechsteinii* (Kuhl, 1817) in Greece. Empty black symbols: published records; filled red symbols: unpublished records.

**Figure 14 animals-13-02529-f014:**
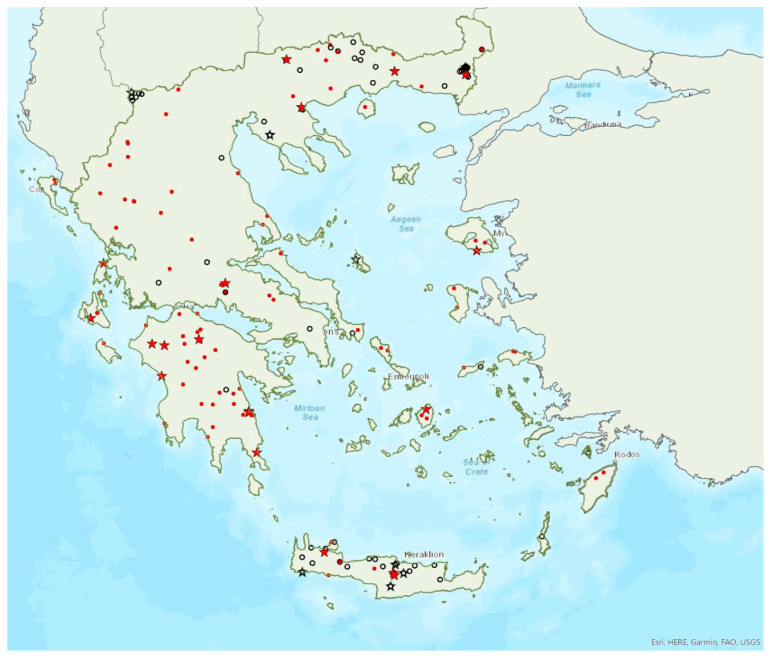
Records of *Myotis emarginatus* (Geoffroy, 1806) in Greece. Empty black symbols: published records. Filled red symbols: unpublished records. Stars: main roosts. Circles: minor roosts and foraging-commuting sites.

**Figure 15 animals-13-02529-f015:**
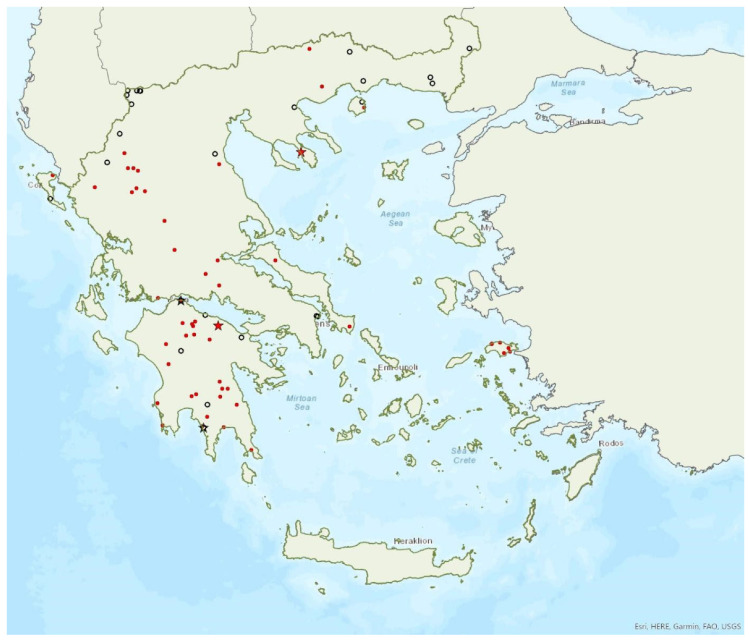
Records of *Myotis nattereri* (Kuhl, 1817) in Greece. Empty black symbols: published records. Filled red symbols: unpublished records. Stars: main roosts. Circles: minor roosts and foraging-commuting sites.

**Figure 16 animals-13-02529-f016:**
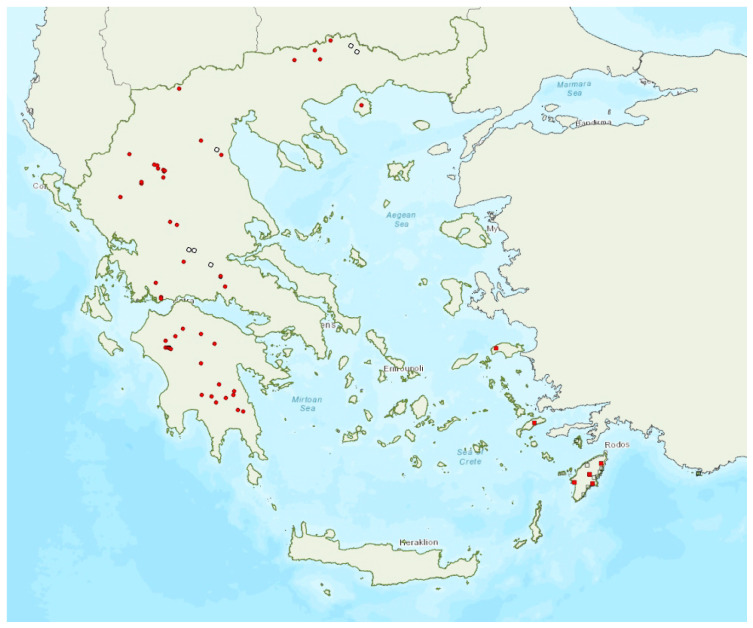
Records of *Eptesicus anatolicus* Felten, 1971 (East Aegean islands, squares) and *Barbastella barbastellus* (Schreber, 1774) (mainland Greece and Thasos island, circles). Empty black symbols: published records; filled red symbols: unpublished records.

**Figure 17 animals-13-02529-f017:**
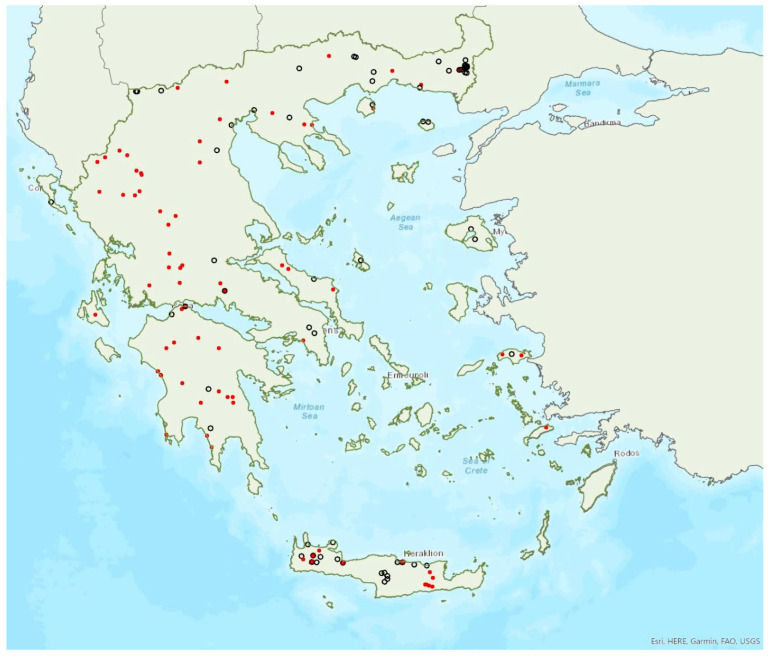
Records of *Eptesicus serotinus* (Schreber, 1774) in Greece. Empty black symbols: published records; filled red symbols: unpublished records.

**Figure 18 animals-13-02529-f018:**
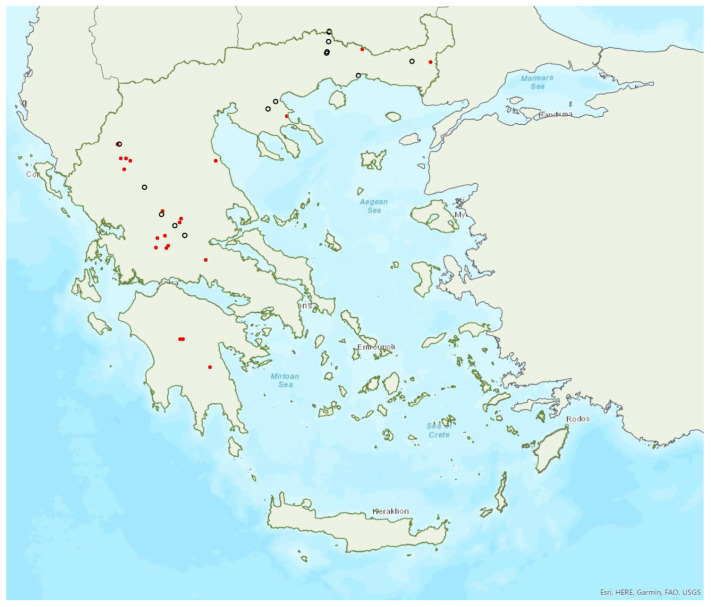
Records of *Nyctalus lasiopterus* (Schreber, 1780) in Greece. Empty black symbols: published records; filled red symbols: unpublished records.

**Figure 19 animals-13-02529-f019:**
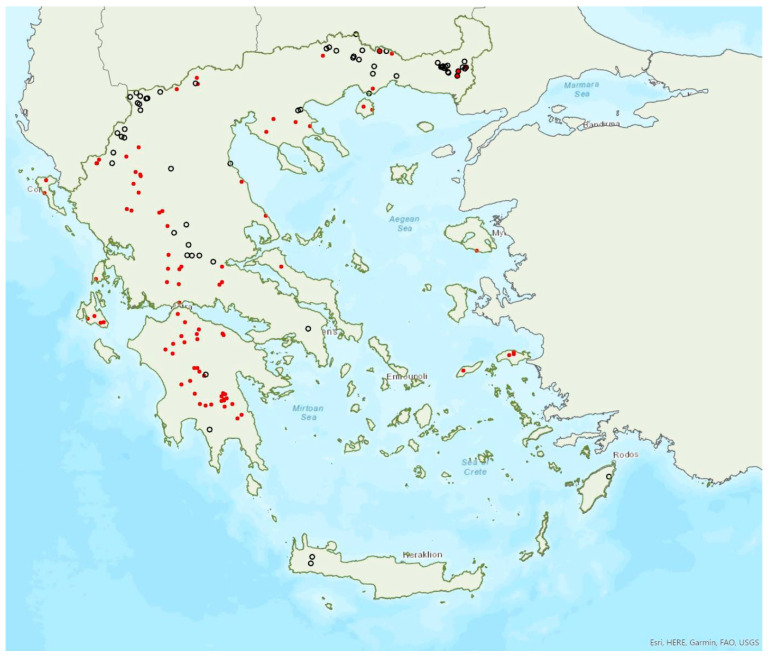
Records of *Nyctalus leisleri* (Kuhl, 1817) in Greece. Empty black symbols: published records; filled red symbols: unpublished records.

**Figure 20 animals-13-02529-f020:**
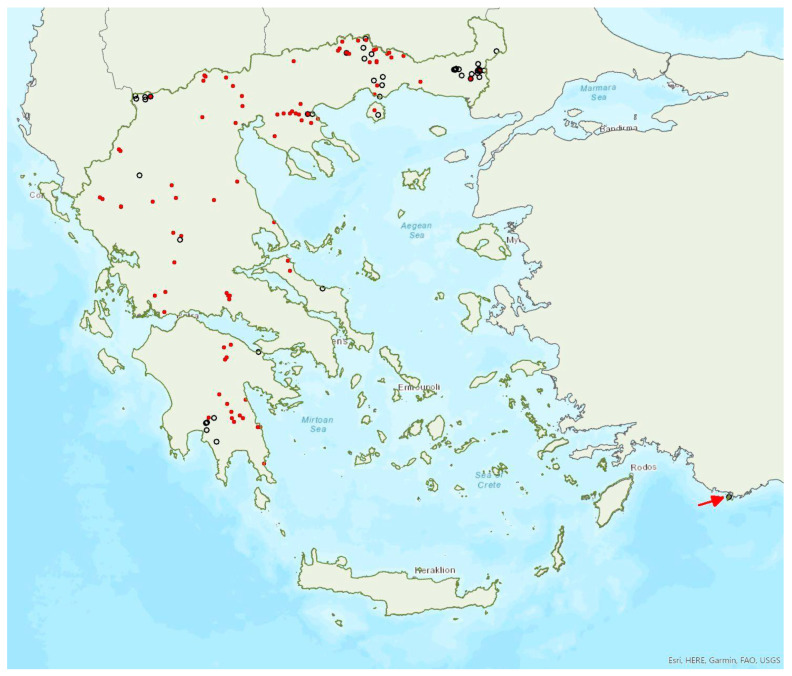
Records of *Nyctalus noctula* (Schreber, 1774) (mainland Greece) and *Rousettus aegyptiacus* (Geoffroy, 1810) (Kastellorizo island, red arrow). Empty black symbols: published records; filled red symbols: unpublished records.

**Figure 21 animals-13-02529-f021:**
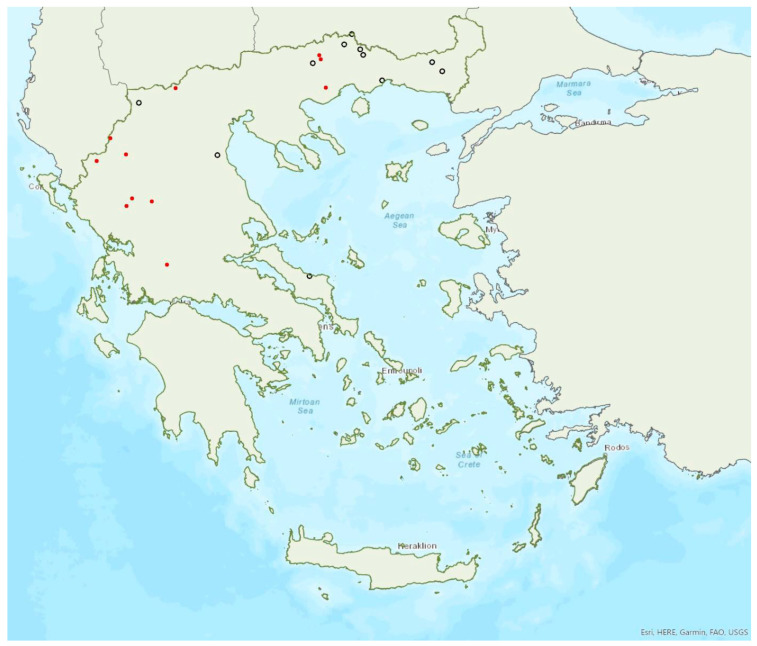
Records of *Vespertilio murinus* Linnaeus, 1758 in Greece. Empty black symbols: published records; filled red symbols: unpublished records.

**Figure 22 animals-13-02529-f022:**
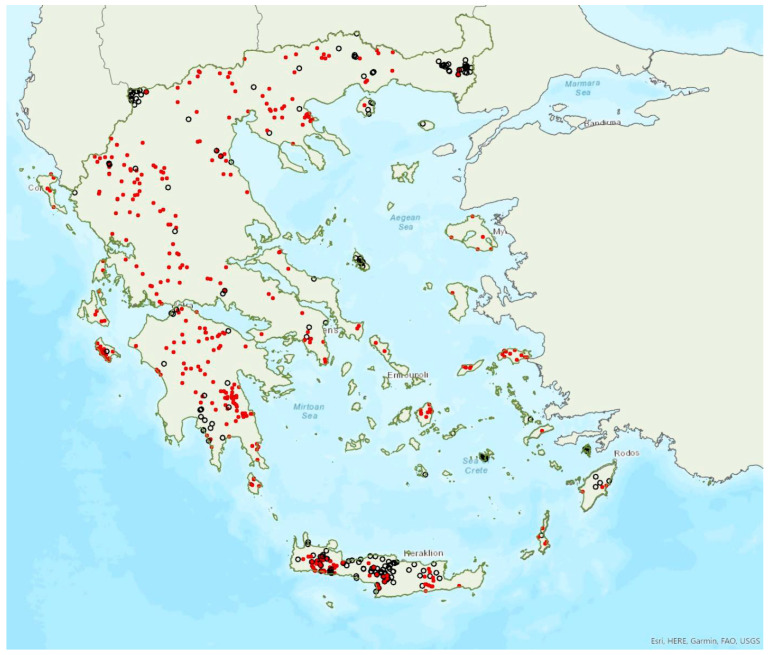
Records of *Hypsugo savii* (Bonaparte, 1837) in Greece. Empty black symbols: published records; filled red symbols: unpublished records.

**Figure 23 animals-13-02529-f023:**
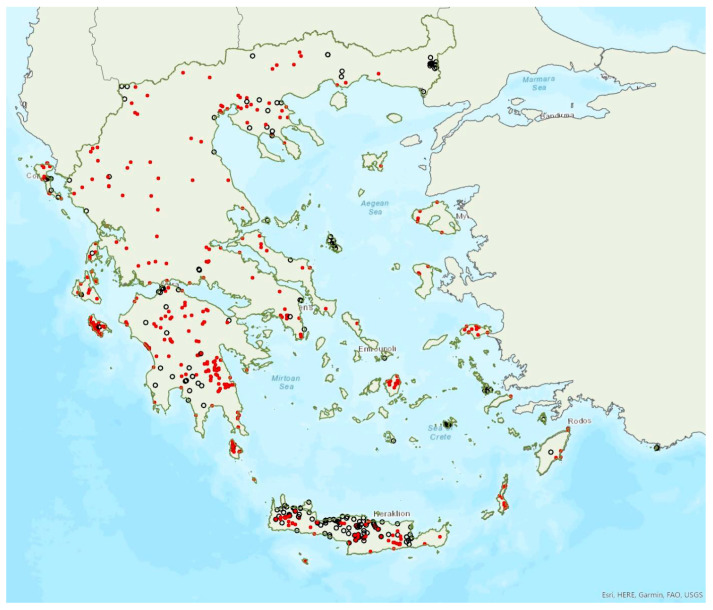
Records of *Pipistrellus kuhlii* (Kuhl, 1817) in Greece. Empty black symbols: published records; filled red symbols: unpublished records.

**Figure 24 animals-13-02529-f024:**
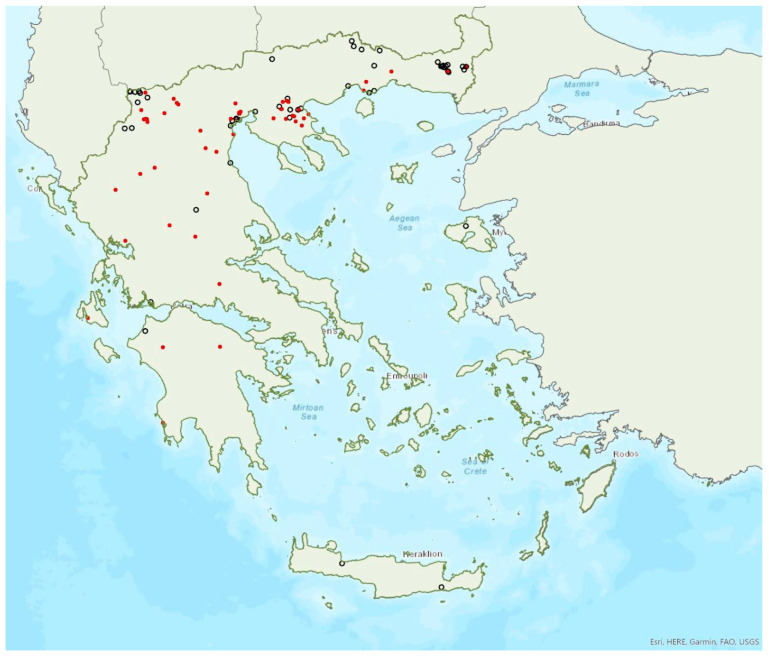
Records of *Pipistrellus nathusii* (von Keyserling et Blasius, 1839) in Greece. Empty black symbols: published records; filled red symbols: unpublished records.

**Figure 25 animals-13-02529-f025:**
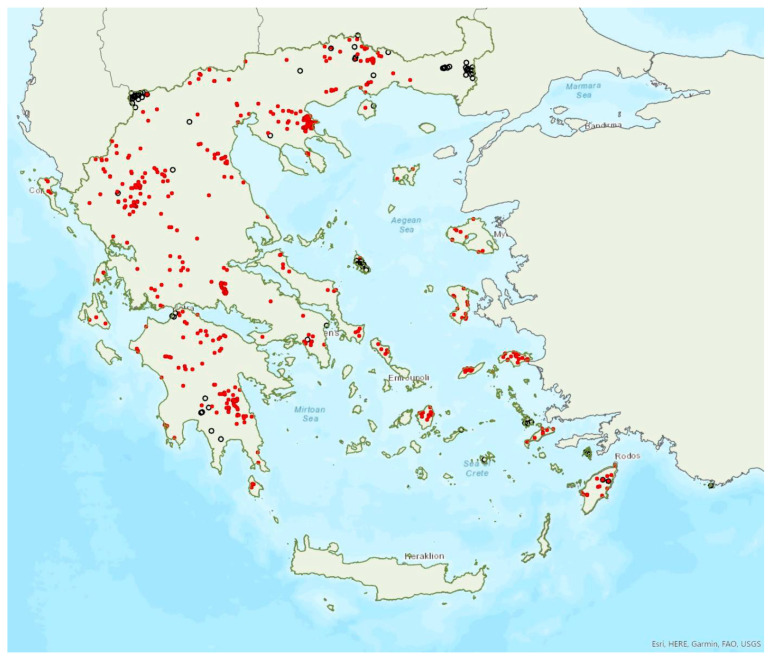
Records of *Pipistrellus pipistrellus* (Schreber, 1774) in Greece. Empty black symbols: published records; filled red symbols: unpublished records.

**Figure 26 animals-13-02529-f026:**
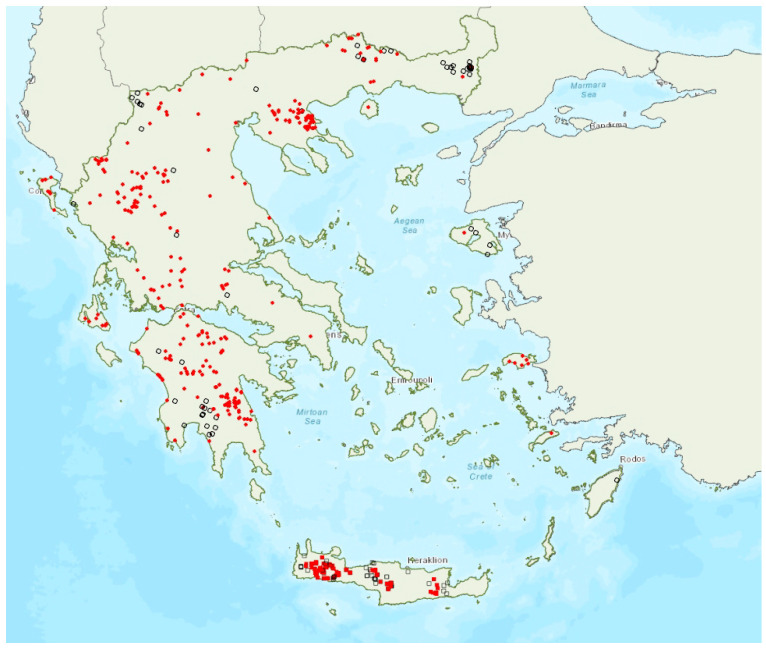
Records of *Pipistrellus pygmaeus* (Leach, 1825) (mainland Greece, Ionian and Aegean islands, circles) and *Pipistrellus hanaki* Hulva et Benda, 2004 (Crete, squares). Empty black symbols: published records; filled red symbols: unpublished records.

**Figure 27 animals-13-02529-f027:**
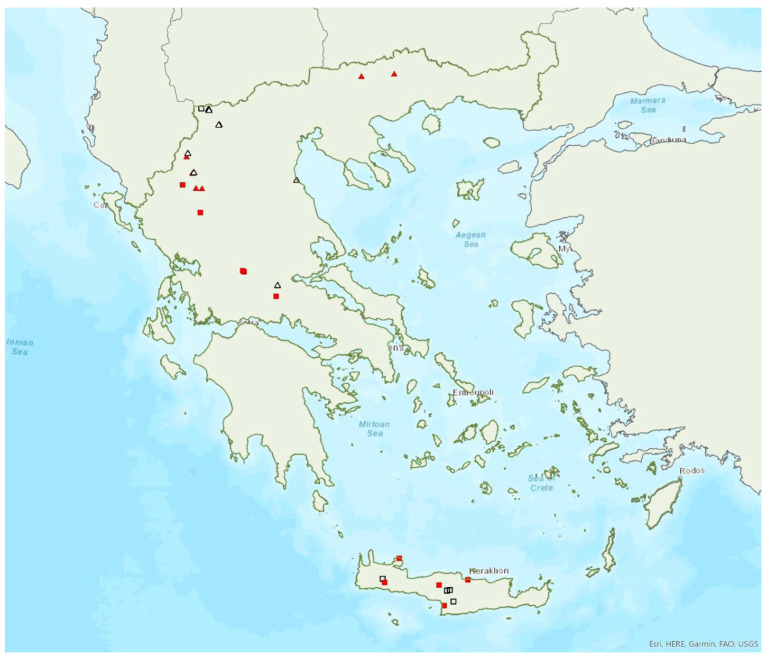
Records of *Plecotus auritus* (Linnaeus, 1758) (triangles) and *Plecotus macrobullaris* Kuzâkin, 1965 (squares) in Greece. Empty black symbols: published records; filled red symbols: unpublished records.

**Figure 28 animals-13-02529-f028:**
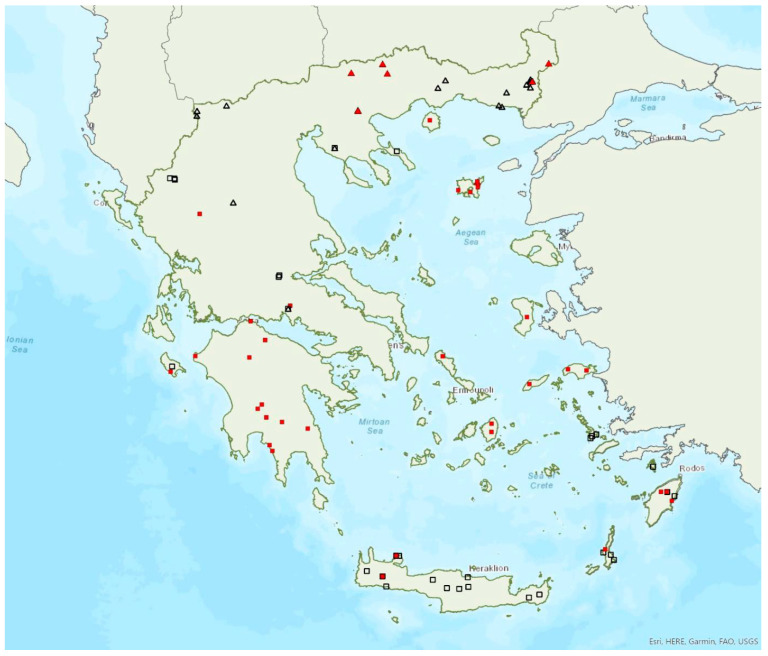
Records of *Plecotus austriacus* (Fischer, 1829) (triangles) and *Plecotus kolombatovici* Đulić, 1980 (squares) in Greece. Empty black symbols: published records; filled red symbols: unpublished records.

**Figure 29 animals-13-02529-f029:**
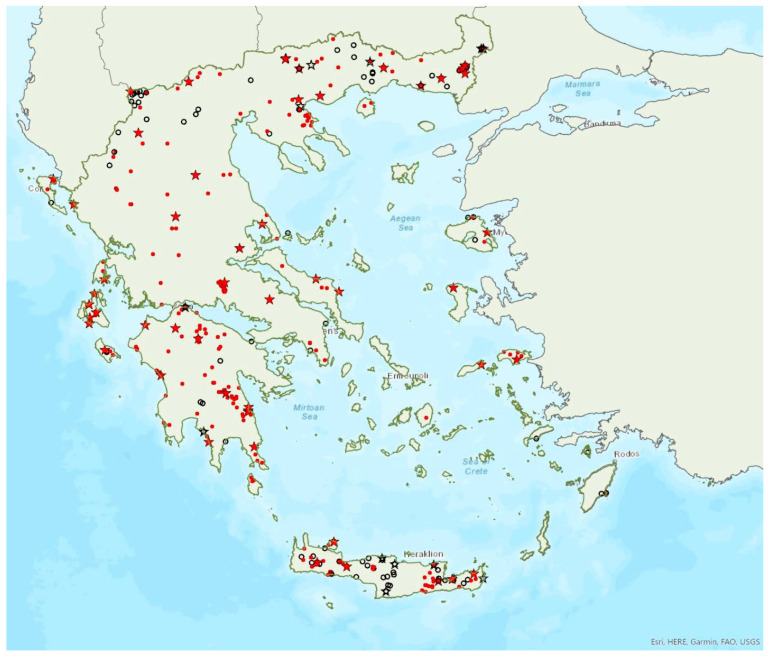
Records of *Miniopterus schreibersii* (Kuhl, 1817) in Greece. Empty black symbols: published records. Filled red symbols: unpublished records. Stars: main roosts. Circles: minor roosts and foraging-commuting sites.

**Figure 30 animals-13-02529-f030:**
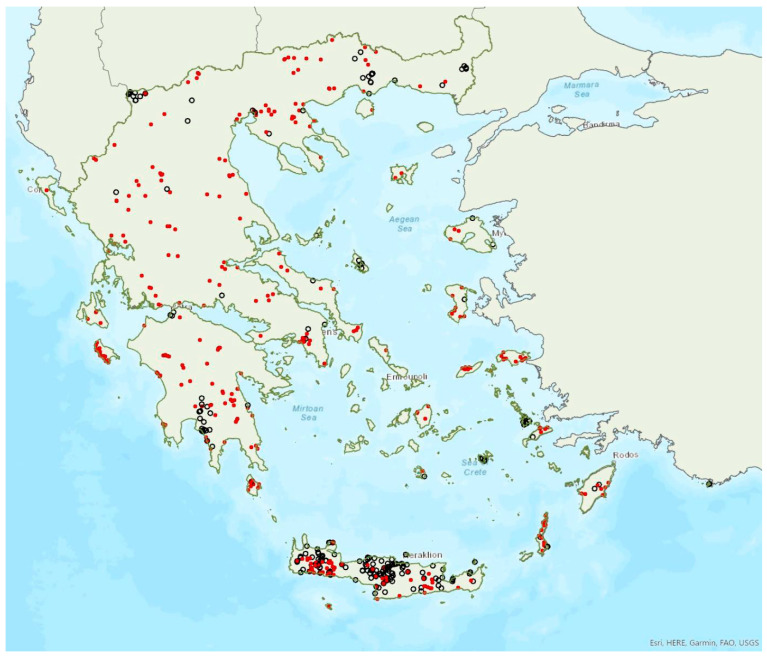
Records of *Tadarida teniotis* (Rafinesque, 1814) in Greece. Empty black symbols: published records; filled red symbols: unpublished records.

**Table 1 animals-13-02529-t001:** Bat species of Greece, their threat category according to the IUCN criteria (in Greece [2] and globally) and their conservation status (Article 17 of the Habitats Directive). Number of occurrence localities in each Region is marked for each species and total number of roosts for the cave dwelling species is provided. Explanations: EN: Endangered, LC: Least Concern, NT: Neat Threatened, DD: Data Deficient, VU: Vulnerable. FV: Favourable, U1: Unfavourable-Inadequate, XX: Unknown, N/A: Not Available.

Species	Red Data Book GR 2009 (IUCN Global)	Conservation Status (Future Prospects)	West Macedonia	Central Macedonia	East Macedonia and Thrace	Epirus	Thessaly	Central Greece	Attica	West Greece	Peloponnese	Ionian Islands	North Aegean	South Aegean	Crete	Roosts
*Rhinolophus ferrumequinum*	LC (LC)	U1 (U1)	28	62	60	16	24	28	5	34	38	29	64	37	119	337
*Rhinolophus hipposideros*	LC (LC)	FV (FV)	44	32	36	19	17	28	13	18	35	26	39	27	113	320
*Rhinolophus euryale*	NT (NT)	U1 (U1)	10	16	24	1	5	3		10	10	11	10	1		77
*Rhinolophus mehelyi*	VU (VU)	XX (XX)		9	9			1	1	6			3			19
*Rhinolophus blasii*	NT (LC)	U1 (U1)	10	20	27	5	5	10	8	10	20	10	21	26	62	120
*Myotis blythii*	LC (LC)	U1 (U1)	14	13	32	9	7	11	2	22	25	10	11	8	55	129
*Myotis myotis*	NT (LC)	U1 (U1)	9	19	18	7	3	16	1	8	6	2	5			55
*Myotis alcathoe*	DD (DD)	XX (XX)	1		6		2	3								
*Myotis davidii*	DD (LC)	XX (XX)	26	25	30	8	9	11	1	11	15	5	6	4	10	
*Myotis brandtii*	DD (LC)	U1 (XX)	2			2										
*Myotis capaccinii*	NT (VU)	U1 (U1)	7	12	32	5	2	2		17	14	12	11		19	45
*Myotis daubentonii*	VU (LC)	XX (XX)	10	6	4								1			
*Myotis bechsteinii*	NT (NT)	U1 (U1)	3	3	8	5	2	3		3	2					
*Myotis emarginatus*	NT (LC)	U1 (U1)	11	12	33	7	6	13	1	15	21	10	9	9	25	82
*Myotis nattereri*	NT (LC)	U1 (U1)	8	4	8	6	4	6	2	12	19	215	6			
*Eptesicus anatolicus*	EN (LC)	U1 (U1)											1	16		
*Eptesicus serotinus*	LC (LC)	XX (FV)	6	10	27	9	5	13	3	12	12	2	5	1	25	
*Nyctalus lasiopterus*	VU (VU)	U1 (U1)	2	3	9	4	7	7			3					
*Nyctalus leisleri*	LC (LC)	U1 (U1)	16	10	34	12	9	12	1	13	26	7	5	1	1	
*Nyctalus noctula*	DD (LC)	U1 (U1)	6	26	45	6	9	8		3	22	5				
*Vespertilio murinus*	DD (LC)	U1 (U1)	2	1	11	5	1	2								
*Hypsugo savii*	LC (LC)	FV (FV)	28	52	50	37	17	31	21	38	86	31	21	38	90	
*Pipistrellus kuhlii*	LC (LC)	FV (FV)	8	41	21	18	8	31	43	50	93	61	20	50	123	
*Pipistrellus nathusii*	DD (LC)	U1 (U1)	22	40	23	3	4	1		3	2	1	1		2	
*Pipistrellus pipistrellus*	DD (LC)	FV (FV)	34	102	66	58	22	52	20	33	75	12	56	55		
*Pipistrellus pygmaeus*	DD (LC)	U1 (U1)	22	74	38	50	16	14	1	48	94	18	11	2		
*Pipistrellus hanaki*	VU (VU)	U1 (U1)													68	
*Barbastella barbastellus*	EN (NT)	U1 (U1)	6	3	6	4	5	7		13	11					
*Plecotus auritus*	VU (LC)	U1 (U1)	8	2	1	3		1								
*Plecotus macrobullaris*	VU (LC)	XX (XX)	1			2		3							7	
*Plecotus austriacus*	DD (NT)	XX (XX)	3	4	13		1	1					1			
*Plecotus kolombatovici*	DD (LC)	XX (XX)		2	1	4		4		4	7	2	9	18	11	
*Miniopterus schreibersii*	NT (VU)	U1 (U1)	18	40	42	12	13	23	9	42	52	21	15	6	63	137
*Tadarida teniotis*	LC (LC)	FV (FV)	16	42	30	17	17	28	25	26	60	20	34	59	137	
*Rousettus aegyptiacus*	- (LC)	N/A (N/A)												1		
No of species			30	29	30	28	26	30	17	24	24	21	24	18	17	

**Table 2 animals-13-02529-t002:** The bat species of Greece and the number of Greek islands known to host each species.

Family	Species	Number of Islands
*Rhinolophidae*	*Rhinolophus ferrumequinum*	22
	*Rhinolophus hipposideros*	19
	*Rhinolophus euryale*	7
	*Rhinolophus mehelyi*	1
	*Rhinolophus blasii*	24
*Vespertilionidae*	*Myotis blythii*	17
	*Myotis myotis*	5
	*Myotis alcathoe*	-
	*Myotis davidii*	10
	*Myotis brandtii*	-
	*Myotis capaccinii*	9
	*Myotis daubentonii*	1
	*Myotis bechsteinii*	1
	*Myotis emarginatus*	18
	*Myotis nattereri*	4
	*Eptesicus anatolicus*	5
	*Eptesicus serotinus*	10
	*Nyctalus lasiopterus*	-
	*Nyctalus leisleri*	10
	*Nyctalus noctula*	4
	*Vespertilio murinus*	1
	*Hypsugo savii*	25
	*Pipistrellus kuhlii*	27
	*Pipistrellus nathusii*	3
	*Pipistrellus pipistrellus*	22
	*Pipistrellus pygmaeus*	7
	*Pipistrellus hanaki*	1
	*Barbastella barbastellus*	1
	*Plecotus auritus*	-
	*Plecotus macrobullaris*	1
	*Plecotus austriacus*	1
	*Plecotus kolombatovici*	13
*Miniopteridae*	*Miniopterus schreibersii*	17
*Molossidae*	*Tadarida teniotis*	27
*Pteropodidae*	*Rousettus aegyptiacus*	1

**Table 3 animals-13-02529-t003:** Altitudinal distribution of bats in Greece.

Species	Mean Altitude of All Records	Median Altitude of Records	Min–Max Altitude of All Records	Min–Max Altitude of Roosts
*Rhinolophus* *ferrumequinum*	373	230	0–1870	0–1500
*Rhinolophus* *hipposideros*	494	400	0–1700	0–1510
*Rhinolophus* *euryale*	341	190	0–1388	0–1388
*Rhinolophus* *mehelyi*	160	80	0–890	0–890
*Rhinolophus* *blasii*	385	277	0–1500	0–1120
*Myotis* *blythii*	361	185	0–2085	0–2085
*Myotis* *myotis*	496	308	5–1950	5–1550
*Myotis* *alcathoe*	608	490	285–930	-
*Myotis* *davidii*	529	425	0–1950	5–1500
*Myotis* *brandtii*	1422	1290	1276–1700	-
*Myotis* *capaccinii*	184	62	0–1120	0–1120
*Myotis* *daubentonii*	502	343	35–1741	250–862
*Myotis* *bechsteinii*	432	320	20–1175	-
*Myotis* *emarginatus*	367	250	0–1380	0–1350
*Myotis* *nattereri*	645	545	0–1870	5–1350
*Eptesicus* *anatolicus*	107	63	5–590	-
*Eptesicus* *serotinus*	474	278	0–1490	45–1490
*Nyctalus* *lasiopterus*	533	417	40–1190	-
*Nyctalus* *leisleri*	623	600	0–2007	280–1050
*Nyctalus* *noctula*	555	425	3–1700	20–180
*Vespertilio* *murinus*	782	890	110–1630	-
*Hypsugo* *savii*	541	465	0–1950	10–1490
*Pipistrellus* *kuhlii*	304	160	0–1440	5–800
*Pipistrellus* *nathusii*	432	370	0–1270	2–640
*Pipistrellus* *pipistrellus*	549	483	0–1870	39–1200
*Pipistrellus* *pygmaeus*	557	580	0–1700	-
*Pipistrellus* *hanaki*	636	545	5–1520	-
*Barbastella* *barbastellus*	884	885	81–1550	1350–1550
*Plecotus* *auritus*	1370	1660	10–1741	-
*Plecotus* *macrobullaris*	974	1013	46–1895	80–1490
*Plecotus* *austriacus*	323	215	40–1120	40–1120
*Plecotus* *kolombatovici*	400	400	5–1490	5–1490
*Miniopterus* *schreibersii*	453	285	0–1500	0–1400
*Tadarida* *teniotis*	443	280	0–1870	862–884
*Rousettus* *aegyptiacus*	25	25	25	-

## Data Availability

The simplyfied bat records included in the Appendix A are also used for the preparation of the Atlas of Greek Mammals, coordinated by the Hellenic Zoological Society. The complete data set is not publicly available due to the sensitivity of bat colonies etc, but more detailed information are available on request from the corresponding author.

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
