# Peer review of "The Bats of Greece: An Updated Review of Their Distribution, Ecology and Conservation"

_animals, 2023, doi:10.3390/ani13152529_

Round 1
Reviewer 1 Report
The manuscript is a thorough review of the species of bats in Greece, with distribution maps and details
of roosts compiled from the literature and adding new records. This paper has the potential to be the go-
to reference manuscript for the country. The manuscript is exceedingly well-written and clear. The main
recommendation I have for improving the paper is to make it clear which data is being published for the
first time and its source. Throughout the species accounts, statements are made that require references,
and if they are data being published for the first time, then there should be more details, such as who
collected the data, when, how, etc. It is unclear which information came from capture records vs.
acoustic monitoring in many sections. I suggest creating sections for each unpublished dataset in the
Supplamental Material and referencing them throughout the manuscript. The paper is long, and I
assume these general statements are an attempt to be succinct, but with this being a blend of review
and new research, it is important to present the new data with the same standards as a research article.
Lastly, the last paragraph of the manuscript is not a conclusion. Please add a summarizing/concluding
paragraph to wrap up the paper.
Here are some other recommendations, edits, and questions:
**Introduction
Ln 40 Always include a comma after “e.g.” and “i.e.” (e.g.,) Edit throughout
Ln 56 Use a en dash (–), rather than a hyphen (-) between ranges. Edit throughout
Ln 76 define NP at first use (ln 70)
Ln 102 – 105 do not capitalize the words in the list.
**Materials & Methods
Will you contribute the occurrence records to the Global Biodiversity Information Facility
(www.gbif.org)?
Ln 132 – 141 End sentence after “reasons,” and start the next sentence with “First, uneven…” remove a,
b, c, make a new sentence at ln 135 with “Second, sampling…” and ln 141 a new sentence with “Third,
the lack…”
Ln 148 Define sea level as 0 m a.s.l. and then just say sea level throughout.
Ln 177 “ranged from 0” m. Always have the unit after the number. Edit throughout
**Results
Ln 158 in the first sentence of each species section, it would be more meaningful to include WHEN the
species was first recorded, rather than by WHO. I recommend removing the names and replacing them
with the year.
Ln 181 – 182 Is it necessary to write out the words and abbreviations every time? Least Concern (LC). If necessary, could you add a table to define all of the Conservation Statuses and abbreviations and use the abbreviations in the text.
Ln 209 “Symbol” should be plural (Symbols). Edit throughout
Ln 215 It would be nice to have the specific islands that are referenced in the species section on that specific distribution map throughout. For example, Lefkada in Fig. 3.
Ln 265 – 272 This paragraph needs references. Where is the data from? Is this “unpublished” data or already published? If it is new data, I suggest including a reference to details of the dataset in the Supplemental Material.
Ln 300 Abbreviate blythii
Ln 302 – 304 Switching tense in the paragraph ln 302 “are” to ln 303 “were.”
Ln 309 Don’t refer to a species as “it,” use the scientific name (and ln 492)
Ln 309 delete most
Ln 328 be consistent with the use of common vs. scientific name throughout the species section. Ln 328 & 338 common name, and ln 332 & 337 scientific name. I suggest using the scientific name throughout and just presenting the common name at the beginning of each species section.
Ln 342 – 343 “The species and roosts…” delete “which is the same for the known roost elevation.” Be succinct.
Ln 357 What is a “plane tree”
Ln 510 Why is it notable that the species in absent in Crete?
Ln 564 use scientific name rather than “serotines” (and on ln 665)
Ln 564 Why are they “all likely breeding females?”
Ln 611 Abbreviate the genus
Ln 637 use scientific name rather than “noctule” (and on ln 665)
Ln 799 use scientific name rather than common pipistrelle
Ln 835 Is 5 m asl sea level or lowland?
Ln 854 use scientific name rather than barbastelles
Ln 930 – 931 P. rather than Pl., like on ln 940
Ln 975, 1024, 1026, 1058, 1060, 1079, 1080 Abbreviate the genus
Ln 1029 – 1034 italicize the scientific names and abbreviate the genus
Table 2 is great!
Ln 1079 What is “True” hibernation?
Ln 1085 17 sea caves
Ln 1089 change babies to pups
Table 3 I suggest moving this table to the Supplemental Material. It is repetitive with what is in the text.
Author Response
We aknowledge reviewer’s effort to read and review our article and we are thankful for the positive comments. Please see the attachment for resposnes to the points raised.

Reviewer 2 Report
The manuscript was well-written. The authors should pay serious attention to cite specific date for a detailed history of bats research in the country.
Author Response
We aknowledge reviewer’s effort to read and review our article and we are thankful for the positive comments. Please see below our resposnes to the points raised.
Point 1: The authors should pay serious attention to cite specific date for a detailed history of bats research in the country.
Reply: The history of bat research in Greece is presented in detail in Hanak et al., 2011 as we state in Line 62. The presentation of a detailed history of bat research in Greece would add a subsequent number of words to the manuscript and it would constitute a repetition of information already presented in this previous publication. In our article we summarise in brief the bat research before 2011 and provide a detailed insight to the more recent advances, this is clearly stated in the text.
Reviewer 3 Report
This papers is acepted and have the potential to be published with minor changes. Most of this changes are related with grammar.
The data of each specie is well explained but not always estandarized.
The figure can be clearer to help the readers to locate each site of Greece. The points and stars used to show the distribution of the species in some cases make it hard to understand when the same color is used, I suggest to use different colors to make it better to the reader. The symbol explanation must be mentioned in each figure to help the readers to understand the symbology without the need to return to the figure 1.

Resolve some comments about english grammar
Author Response
We aknowledge reviewer’s effort to read and review our article and we are thankful for the positive comments. Please see below our resposnes to the points raised.
Regarding species data standarization, we tried to be consice but not repetitive, although another author stressed the repetitive nature of species accounts.
Regarding the maps, we followed the reviewers' suggestion to use differrent colours and provide explanations in each map.
Please see the attachment for resposnes to some specific points raised in the peer-review-30133255.v1.pdf file.
Please have in mind that many of the co-authors of this article are native english spekers with long-lasting experience in writing scientific documents. Therefore, as stated from other reviewers, there is no need for a thourough rewriting of the article.

Reviewer 4 Report
Manuscript ID animals-2458148
The bats of Greece: an updated review of their distribution, ecology and conservation
Section Mammals
Special Issue Mammal Ecology and Conservation in Southeastern Europe
This ms reports the basic occurrence of bats in Greece, based both on existing literature, and on very many unpublished records. Each species account is written on the same scheme (hence a bit repetitive), but the exact source upon which comments are based are unknown or somehow detailed in a supplementary file. Regarding the repetitive nature of the ms, for instance, the sentence “On warm and calm winter nights bats may emerge to forage” appears for many species, but is it based on verifiable records for all these species, or just based in the authors’ impression? Adding whether these remarks were based on e.g. captures or acoustic observation would help further readers to appreciate how reliable such remarks are. Records are then mapped on a map of Greece.
Overall, the ms brings much needed new information on the bats of Greece, where habitat and land use changes by humans put a huge pressure on the conservation of these mammals.
Introduction: one of the main aims announced in the Intro (c) is to highlight their conservation needs. Unfortunately, I see nowhere how this was established. Therefore, I don’t really understand the meaning of the last sentence appearing at the end of the account of every species. This is certainly important information, but it should be explained in the M & M section, minimally the meaning of the different categories used and perhaps also how you reach to such conclusions. IUCN criteria are more universally known.
M & M : apparently, you did not try to include data from e.g. iNaturalist or GBIF. This is a pity since few records can be easily verified on the provided pictures and may concern rare species (e.g. E. bottae).
L187: I am surprized that you use (André, 1797) as the authority for Rh. hipposideros; usually, we refer to it as (Bechstein, 1800), see e.g. IT IS, UICN, GBIF, etc. Unless there was a recent taxonomic change which I am unaware, such changes should not be used here and Bechstein kept.
L240-243: this sentence seems to miss few words/verbs. Are you referring only to records on Levos? This is not clear. On L245 it seems that it becomes again general. Please revise.
L301: to my knowledge, the oxygnathus taxon (whether species or subspecies) was never recorded from the Middle-East or the Caucasus, which is exclusively inhabited by omari see e.g. ref. 82 or 88.
L309: remove “most”.
L312/L342: are you sure to have found a roost at 2085 m a.sl. a cave in Mt Olympic? As this might be a record, it would be informative to detail a little bit better this extreme observation.
L453: in sympatry
L384, 385: a coma is missing after descriptors and before 2004
L873: replace Benda et al, 2019 by a numbered version
L1024: regarding the occurrence of M. mystacinus in Greece, it seems important to stress again in this discussion that the species was once considered as common and widespread, but because of recent genetic work, most records are now assigned to M. davidii. Otherwise, thisseems a bit odd.
L. 1029 and beyond: most italics to species names are missing.
Literature: the authors used caps for the first letter of most words, which is not correct and should be revised carefully to match usual English spelling, or even German spelling (e.g. L1410).
Well written, and concise. No major change required, except for the small spellings/mistakes mentioned above.
Author Response

(The authors gave the same response as above.)

Reviewer 5 Report
This manuscript summarizes the current knowledge of bats in Greece and provides updated distribution maps. Such an effort is timely and much needed. However, I think the manuscript can be improved by a few more summary tables/figures/maps.
First, a map of Greece with general regions (such as locations mentioned in the introduction) labelled can really help any international readers.
Second, Table 1 (or a separate table) should include family level taxonomic information. Additionally the top row of Table 1 is not displaying on my end.
Third, I highly recommend authors making a simple comparison figure to show conservation status differences between national and IUCN levels. I noticed that at least a few species (such as Plecotus macrobullaris) are recognized as vulnerable by IUCN but still unknow in the national legislation. I think authors should highlight such information/legal protection deficiencies, which should be used as the leverage to argue for more research funding in the future.
Fourth, since ecology is part of the article, I think it will be extremely helpful to have one summary table to describe each species’ diet. This information can be used to demonstrate how many of them potentially provide ecosystem services such as pest control (insectivores) or pollination (nectarivores).
Fifth, given the long length of this article, a table of content with page numbers/page links might be helpful. I think some readers will want to jump to a specific species instead of reading through the whole paper. I don’t know if it is possible but maybe editors from MDPI can help.
Author Response
We aknowledge reviewer’s effort to read and review our article and we are thankful for the positive comments. Please see below our resposnes to the points raised.
This manuscript summarizes the current knowledge of bats in Greece and provides updated distribution maps. Such an effort is timely and much needed. However, I think the manuscript can be improved by a few more summary tables/figures/maps.
First, a map of Greece with general regions (such as locations mentioned in the introduction) labelled can really help any international readers.
Suggestion followed. A map of Greece as required has now been added.
Second, Table 1 (or a separate table) should include family level taxonomic information. Additionally the top row of Table 1 is not displaying on my end.
Suggestion followed. This information was added in Table 2, which is smaller than Table 1, for clarity. The editorial team can decide if it has to be added elsewhere.
Third, I highly recommend authors making a simple comparison figure to show conservation status differences between national and IUCN levels. I noticed that at least a few species (such as Plecotus macrobullaris) are recognized as vulnerable by IUCN but still unknow in the national legislation. I think authors should highlight such information/legal protection deficiencies, which should be used as the leverage to argue for more research funding in the future.
The global listing is now included in Table 1, together with the Greek one. Differences in IUCN classification are explained by many factors, including number of localities, EOO, AOO which make comparisons interesting but also difficult and lengthy.
Fourth, since ecology is part of the article, I think it will be extremely helpful to have one summary table to describe each species’ diet. This information can be used to demonstrate how many of them potentially provide ecosystem services such as pest control (insectivores) or pollination (nectarivores).
This is a good suggestion but unfortunately there are no data on bat species’ diet in Greece. For dietary preferences of European bats the reader can refer to the following publications:
Dietz, C.; von Helversen, O.; Nill, D. Bats of Britain, Europe and Northwest Africa; A & C Black Publishers Ltd.: London, UK, 2009; ISBN 1-4081-0531-4.
and
Dietz, C.; Kiefer, A. Bats of Britain and Europe; Bloomsbury Publishing Plc: London, UK, 2016;
Fifth, given the long length of this article, a table of content with page numbers/page links might be helpful. I think some readers will want to jump to a specific species instead of reading through the whole paper. I don’t know if it is possible but maybe editors from MDPI can help.
The editorial team can decide if this is possible. If yes, we totally agree.